# Disruption of the PIKfyve complex unveils an adaptive mechanism to promote lysosomal repair and mitochondrial homeostasis

Candice Kutchukian ⓘ, Maria Casas ⓘ, Rose E. Dixon ⓘ & Eamonn J. Dickson ⓘ ✉

Lysosomes are essential organelles that regulate cellular homeostasis through complex membrane interactions. Phosphoinositide lipids play critical roles in orchestrating these functions by recruiting specific proteins to organelle membranes. The PIKfyve/Fig4/Vac14 complex regulates $PI(3,5)P_2$ metabolism, and intriguingly, while loss-of-function mutations cause neurodegeneration, acute PIKfyve inhibition shows therapeutic potential in neurodegenerative disorders. We demonstrate that PIKfyve/Fig4/Vac14 dysfunction triggers a compensatory response where reduced mTORC1 activity leads to ULK1-dependent trafficking of ATG9A and PI4KIIα from the TGN to lysosomes. This increases lysosomal PI(4)P, facilitating cholesterol and phosphatidylserine transport at ER-lysosome contacts to promote membrane repair. Concurrently, elevated lysosomal PI(4)P recruits ORP1L to ER-lysosome-mitochondria three-way contacts, enabling PI(4)P transfer to mitochondria that drives ULK1-dependent fragmentation and increased respiration. These findings reveal a role for PIKfyve/Fig4/Vac14 in coordinating lysosomal repair and mitochondrial homeostasis, offering insights into cellular stress responses.

Lysosomes play a central role in cellular catabolism, as their acidic interior creates an optimal environment for >50 different hydrolases to break down and recycle cellular waste. Beyond this well-established role, lysosomes also serve as crucial regulators of various cellular functions, including autophagy and lipid metabolism[1]. The preservation of lysosomal function and structural integrity is crucial for maintaining normal cellular physiology, as underscored by the wide range of diseases associated with lysosomal dysfunction, such as lysosomal storage diseases, neurodegenerative disorders, autoimmune diseases, and cancer[2–5]. Notably, a growing number of studies highlight the significance of phosphoinositides (PIs) and their metabolizing enzymes in modulating lysosomal function, dynamics, and homeostasis[6,7].

Although minor in membrane composition, PIs play a pivotal role in a diverse range of cellular processes including membrane trafficking, ion channel regulation, cytoskeletal reorganization, and autophagy[8,9]. Specifically, PIs encompass seven distinct species that

can be interconverted through phosphorylation/dephosphorylation at the D3, D4, and D5 positions of the inositol headgroup. These enzymatic reactions are mediated by lipid PI-kinases and phosphatases, whose dynamic subcellular localization results in a unique and convertible PIs signature among different membrane compartments.

At the lysosomal surface, the PIP 5-kinase, PIKfyve, and its antagonist $PIP_2$ 5-phosphatase, Fig4, form a ternary complex together with the scaffold protein Vac14 to ensure the synthesis and interconversion of the low-abundance PIs, phosphatidylinositol 3-phosphate (PI(3)P) and phosphatidylinositol 3,5-biphosphate ($PI(3,5)P_2$; Fig. 1a)[10–12]. At lysosomal membranes, the PIKfyve/Fig4/Vac14 complex (referred hereafter as the PIKfyve complex) is strongly interdependent and interconnected on each of the triad members, this is underscored by Fig4 loss-of-function in the *pale tremor* mouse which results in decreased $PI(3,5)P_2$ levels[13], reflecting its steady-state role in stimulating PIKfyve kinase activity. Similarly, fibroblasts from the Vac14-null mice also show a reduction in $PI(3,5)P_2$ levels[14], again

Department of Physiology and Membrane Biology, University of California, Davis, CA 95616, USA. ✉e-mail: ejdickson@health.ucdavis.edu

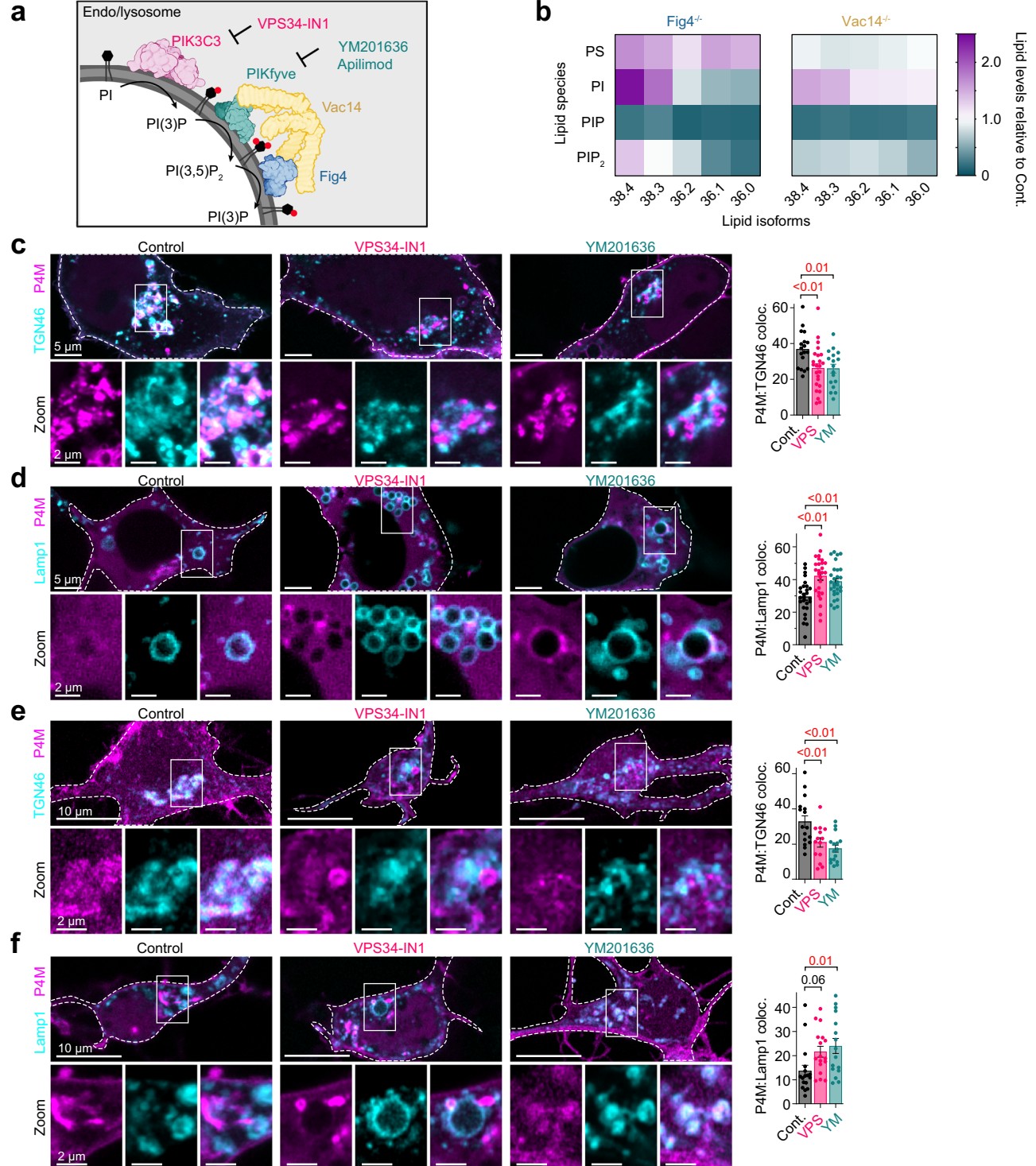

**Fig. 1 | PIKfyve complex controls TGN and lysosomal PI(4)P levels. a** Diagram depicting PI(3,5)P$_2$ regulation by the PIKfyve complex at lysosomal membranes. *Created in BioRender. Kutchukian, C. (2025)* https://BioRender.com/cn6rz3y. **b** Heatmap showing the abundance of PS, PI, PIP, and PIP$_2$ lipid isoforms in Fig4- and Vac14-deficient cells, relative to control. **c** Left: Confocal images of HEK293t cells expressing P4M-YFP (magenta) and TGN46-mCherry (cyan), and treated with either DMSO (control), VPS34-IN1- or YM201636. Right: Quantification of P4M area colocalizing within TGN46 signal (%). Control: $n = 18$ cells; VPS34-IN1: $n = 24$;

YM201636: $n = 17$. **d** Left: Same as (**c**) only Lamp1-GFP expressed instead of TGN46-mCherry. Right: Quantification of P4M area within Lamp1 area. Control: $n = 27$ cells; VPS34-IN1: $n = 27$; YM201636: $n = 29$. **e, f** Same as c and d, only in cortical neurons co-expressing P4M-YFP with either TGN46-mCherry ((**e**) Control: $n = 16$ neurons; VPS34-IN1: $n = 14$; YM201636: $n = 16$) or Lamp1-RFP ((**f**) Control: $n = 17$ neurons; VPS34-IN1: $n = 17$; YM201636: $n = 16$). For (**c–f**) data are presented as mean ± s.e.m. Statistical significance was determined using ordinary one-way ANOVA.

underscoring the importance of the PIKfyve complex in maintaining steady-state PI(3,5)P$_2$ levels. Finally, the significance of this complex is highlighted in humans where loss-of-function mutations in *FIG4* and *VAC14* genes result in severe neurological disorders including

Amyotrophic Lateral Sclerosis (ALS), Charcot Marie Tooth Type 4 J and Yunis-Varón Syndrome[13,15–18].

Despite the clear necessity for proper PIKfyve function, pharmacological inhibition of PIKfyve, which like disease mutations also

decreases PI(3,5)P$_2$ levels[19], has emerged as a promising therapeutic approach for neurodegeneration. Recent studies have demonstrated that PIKfyve inhibitors can mitigate disease progression in diverse ALS models[20], suggesting a biphasic relationship between PIKfyve activity and neuronal survival. This apparent contradiction, where both genetic dysfunction and pharmacological inhibition of the same pathway can influence neurodegeneration, albeit in opposite directions, presents a mechanistic puzzle. Resolving this paradox could provide crucial insights into fundamental cellular adaptation mechanisms and potentially unveil novel therapeutic targets for a spectrum of neurodegenerative conditions characterized by lysosomal dysfunction.

In this current study, we determine that disruption of the PIKfyve complex leads to the production of PI(4)P on lysosomal membranes, a process that involves the redistribution of PI(4)P-synthesizing enzyme PI4KIIα from the TGN to the lysosome together with the integral membrane protein ATG9A. Our findings indicate that this shift is initiated by a small reduction in mTOR activity which releases an inhibitory break on ULK1-dependent trafficking of PI4KIIα and ATG9A from the TGN to the lysosomal membranes. The consequent elevation in lysosomal PI(4)P serves dual functions: it enhances membrane repair through OSBP/ORP-mediated delivery of phosphatidylserine and cholesterol, while simultaneously facilitating PI(4)P transfer at ER-lysosome-mitochondria contacts to drive mitochondrial fragmentation and increased respiratory capacity. Pharmacological inhibition of either PI4KIIα or ULK1 reverses these phenotypes, restoring normal lysosomal and mitochondrial homeostasis. Our findings reveal regulatory roles for the PIKfyve complex in coordinating lysosomal membrane repair and mitochondrial function through spatiotemporal control of PI(4)P, illuminating the intricate interplay between phosphoinositide metabolism, membrane dynamics, and organellar crosstalk with important implications for neurodegenerative disorders.

## Results

### PIKfyve complex influences PI(4)P distribution

At endosomal membranes, the PIKfyve complex controls the synthesis and turnover of the low abundant phosphoinositide PI(3,5)P$_2$ (Fig. 1a), with knockout of PIKfyve[21] or loss of function mutations in *FIG4* and *VAC14* resulting in a significant drop in cellular PI(3,5)P$_2$ levels[13,14,22]. To comprehensively examine how PIKfyve complex disruption affects phosphoinositide distribution, we employed complementary approaches targeting different aspects of phosphoinositide biology. Using ultra-high pressure liquid chromatography coupled to tandem mass spectrometry (UPLC-MS/MS), we quantified total cellular phosphoinositide levels from control or CRISPR-edited, FIG4 knockout (Fig4$^{-/-}$) or VAC14 knockout (Vac14$^{-/-}$) cells[23]. Among all phosphoinositide lipid species analyzed, phosphatidylinositol monophosphate (PIP) showed significant alterations, with average levels being decreased in both Fig4$^{-/-}$ and Vac14$^{-/-}$ cells relative to control cells (Fig. 1b; Supplementary Fig. 1a). It is important to note that UPLC-MS/MS cannot differentiate between PIP regioisomers (PI(3)P, PI(4)P, PI(5)P), so these measurements represent bulk changes in all monophosphorylated phosphoinositides. Given that PI(4)P is the most abundant PIP species (-10-fold >PI(3)P)[9,24], the observed changes likely reflect primarily PI(4)P alterations, though contributions from other PIP species cannot be excluded. To complement our lipidomic approaches we expressed the PI(4)P fluorescent biosensor P4M-YFP[25] in control, Fig4$^{-/-}$ and Vac14$^{-/-}$ cells. In control cells, P4M-YFP exhibited a prominent trans-Golgi network (TGN) distribution as expected (Supplementary Fig. 1b), where PI(4)P is synthesized by PI4KIIα and PI4KIIIβ[26]. Strikingly, Fig4$^{-/-}$ and Vac14$^{-/-}$ cells displayed a significant decrease of P4M-YFP at TGN regions, with a corresponding accumulation of the biosensor on internal membranes (Supplementary Fig. 1b). We also observed a significant decrease in plasma membrane PI(4)P levels (Supplementary Fig. 1b), suggesting a global redistribution of cellular PI(4)P pools.

To validate these findings and extend them to physiologically relevant cell types, we employed a pharmacological approach. We targeted two key kinases involved in PI(3,5)P$_2$ synthesis: phosphatidylinositol 3-kinase catalytic subunit type 3 (PIK3C3, also known as VPS34) using VPS34-IN1[27], and PIKfyve using YM201636[28] (Fig. 1a). We first confirmed the efficacy of these inhibitors using established biosensors for PI(3)P (GFP-2xFyve[29]) and PI(3,5)P$_2$ (GFP-2xPX[19]) (Supplementary Fig. 1c, d). Consistent with our genetic models, pharmacological inhibition of PI(3,5)P$_2$ synthesis significantly reduced PI(4)P levels at TGN membranes (Fig. 1c) while enhancing PI(4)P accumulation on lysosomal membranes (Fig. 1d, Supplementary Fig. 1b). These effects occurred without discernible changes in TGN structure (Supplementary Fig. 1e), though lysosomes appeared enlarged (Supplementary Fig. 1f), consistent with the well-documented vacuolization effect of PIKfyve inhibition[30-32]. Importantly, similar PI(4)P redistribution was observed with acute (2 h) treatment of cells with the selective PIKfyve inhibitor Apilimod (Supplementary Fig. 1g), indicating these effects represent direct consequences of PIKfyve inhibition rather than secondary adaptations. Identical results were observed in neurons with decreased TGN PI(4)P (Fig. 1e) and increased lysosomal PI(4)P levels (Fig. 1f). Collectively, these findings reveal a previously unappreciated role for the PIKfyve complex in regulating the subcellular distribution of PI(4)P.

### PIKfyve complex disruption induces a decrease of PI4KIIα on TGN membranes

To elucidate the mechanistic basis for TGN PI(4)P reduction upon PIKfyve complex disruption, we systematically investigated the enzymes regulating PI(4)P homeostasis at the TGN (Fig. 2a). At TGN membranes, PI(4)P is generated from PI by two distinct PI 4-kinase isoforms, PI4KIIα and PI4KIIIβ[33-35]. Western blot analysis revealed that total protein levels of both kinases remained unchanged in Fig4$^{-/-}$ and Vac14$^{-/-}$ cells compared to controls (Supplementary Fig. 2a), indicating that any alterations in PI(4)P levels likely resulted from changes in enzyme localization rather than expression.

Immunofluorescence analysis of subcellular distribution using anti-PI4KIIα and anti-PI4KIIIβ antibodies co-stained with the TGN marker TGN46 revealed striking differences between these kinases. While PI4KIIIβ maintained its characteristic TGN enrichment in Fig4$^{-/-}$ and Vac14$^{-/-}$ cells (Supplementary Fig. 2b), PI4KIIα immunolabeling appeared significantly more punctate and dispersed throughout the cytosol, with quantitative analysis confirming a marked reduction in its TGN association (Fig. 2b). This PI4KIIα redistribution was recapitulated in primary cortical neurons treated with VPS34-IN1 or YM201636 (Fig. 2c), establishing this phenomenon as a direct consequence of PIKfyve complex disruption across multiple cell types.

Since PI4KIIα localization at TGN membranes depends on its palmitoylation by DHHC3[36], we investigated whether altered DHHC3 distribution might contribute to PI4KIIα redistribution. Indeed, immunofluorescence analysis revealed that endogenous DHHC3, normally enriched in TGN regions in control cells, appeared significantly more dispersed in Fig4$^{-/-}$ and Vac14$^{-/-}$ cells, with quantitative measurements confirming reduced DHHC3 fluorescence intensity (Fig. 2d).

Together, these results reveal that PIKfyve complex disruption leads to a reduction of TGN PI(4)P, likely through DHHC3-dependent redistribution of PI4KIIα. The decreased association of PI4KIIα with TGN membranes diminishes local PI(4)P synthesis capacity, effectively reshaping the cellular PI(4)P landscape.

### PIKfyve complex disruption increases PI4KIIα trafficking to lysosomes

Given the observed depletion of PI4KIIα from TGN membranes, we hypothesized that PIKfyve complex disruption redirects PI4KIIα to lysosomal compartments (Fig. 3a), thereby accounting for the

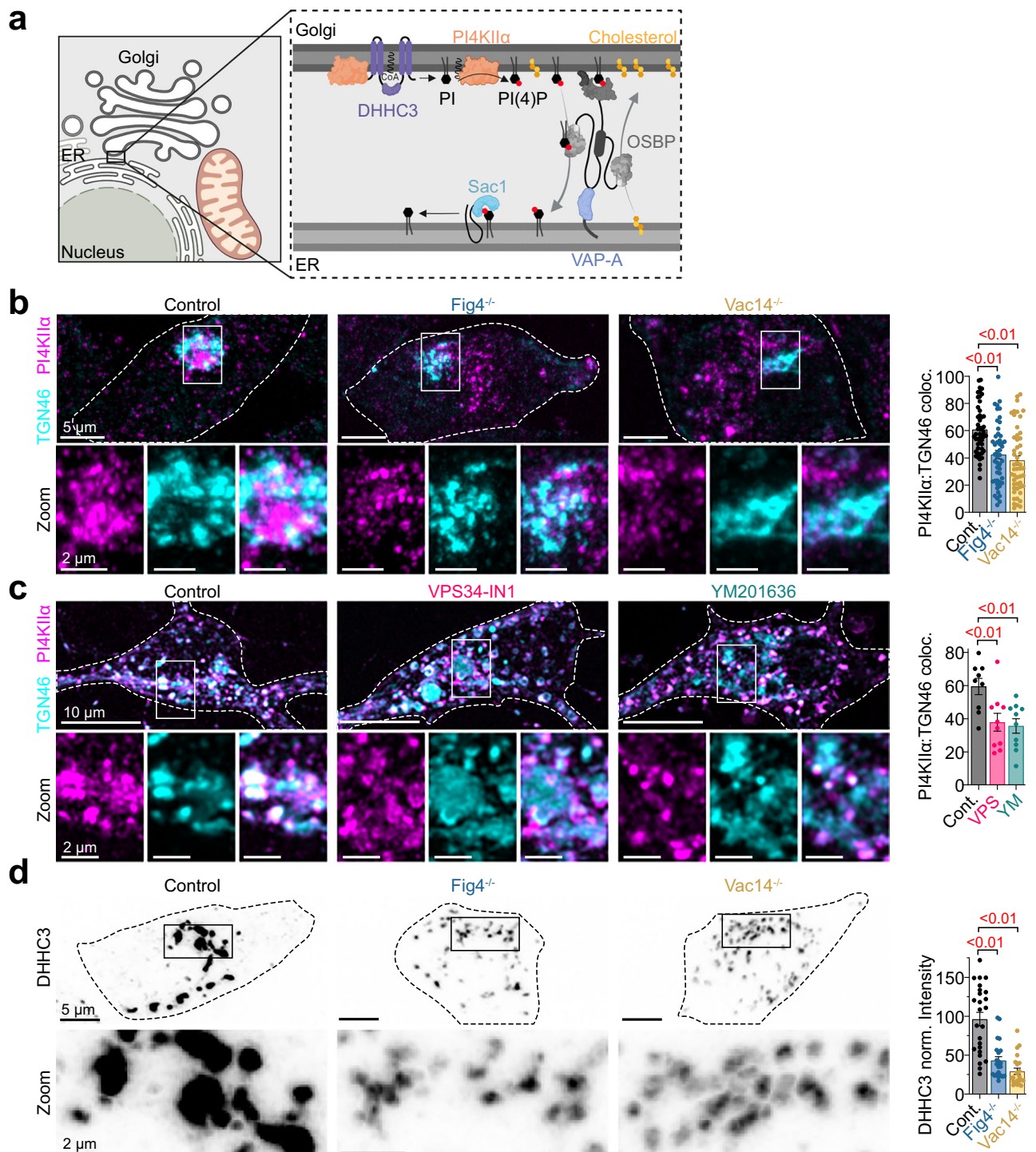

**Fig. 2 | Loss of PIKfyve complex function reduces PI4KIIα at the TGN.**
**a** Schematic illustrating PI(4)P regulation at ER-TGN MCS. *Created in BioRender.*
*Kutchukian, C. (2025)* https://BioRender.com/vfa0ad6. **b** Left: Confocal images of
control, Fig4$^{-/-}$ and Vac14$^{-/-}$ cells fixed and stained for PI4KIIα (magenta) and
TGN46 (cyan). The bottom row shows magnified views of PI4KIIα in TGN regions.
Right: Quantification of PI4KIIα area within TGN46 signal (%). Control: $n = 50$ cells;
Fig4$^{-/-}$: $n = 49$; Vac14$^{-/-}$: $n = 50$. **c** Left: Same as (**b**) only in cortical neurons express-
ing PI4KIIα-GFP and TGN46-mCherry and treated with DMSO (control), VPS34-IN1

or YM201636. Right: Quantification of PI4KIIα area within TGN46 area (%). Control:
$n = 9$ neurons; VPS34-IN1: $n = 10$; YM201636: $n = 10$. **d** Left: Confocal images of
control, Fig4$^{-/-}$ and Vac14$^{-/-}$ cells fixed and immunostained with anti-DHHC3. The
bottom row shows expanded views of DHHC3 in the perinuclear area. Right:
Quantification of DHHC3 intensity normalized to the cytosolic intensity. Control:
$n = 25$ cells; Fig4$^{-/-}$: $n = 22$; Vac14$^{-/-}$: $n = 22$. All data are presented as mean ± s.e.m.
For (**b**−**d**) statistical significance was determined using ordinary one-way ANOVA.

increased PI(4)P on Lamp1-positive membranes (Fig. 1d, f). Immuno-
fluorescence analysis of Fig4$^{-/-}$ and Vac14$^{-/-}$ cells co-stained with anti-
PI4KIIα and anti-Lamp1 antibodies revealed a significant increase in
PI4KIIα-Lamp1 colocalization in Fig4$^{-/-}$ cells, with a similar trend in
Vac14$^{-/-}$ cells (Fig. 3b). This redistribution phenomenon was

reproduced in cortical neurons expressing PI4KIIα-GFP and Lamp1-
RFP following treatment with VPS34-IN1 or YM201636 (Fig. 3c), or
HEK293t cells transiently treated with Apilimod (Supplementary
Fig. 3a), confirming conservation of this mechanism across different
cell types.

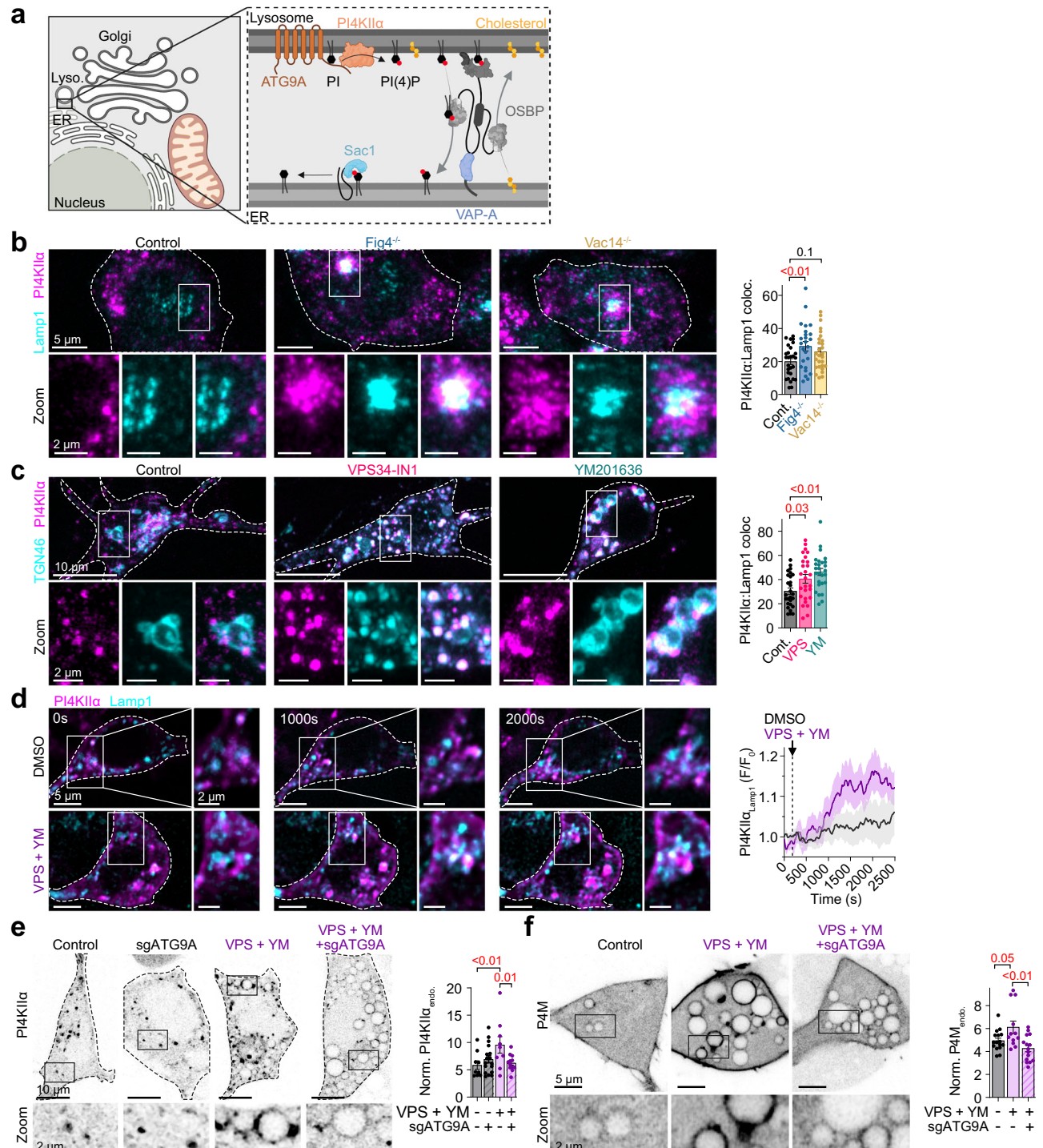

**Fig. 3 | Loss of PIKfyve complex function increases PI4KIIα on lysosome membranes. a** Schematic illustrating PI(4)P regulation at ER-Lysosome MCS. *Created in BioRender. Kutchukian, C. (2025)* https://BioRender.com/2j6x99o. **b** Left: Confocal images of control, Fig4[−/−] and Vac14[−/−] cells fixed and immunolabeled for PI4KIIα (magenta) and Lamp1 (cyan). The bottom row shows magnified views of PI4KIIα in lysosomal regions. Right: Quantification of PI4KIIα area within Lamp1 signal (%). Control: *n* = 28 cells; Fig4[−/−]: *n* = 25; Vac14[−/−]: *n* = 28. **c** Left: cortical neurons expressing PI4KIIα-GFP and Lamp1-RFP and treated with DMSO (control), VPS34-IN1 or YM201636. Right: Quantification of PI4KIIα area within Lamp1 area (%) (*n* = 27 neurons for each condition). **d** Left: Representative confocal images of PI4KIIα-GFP and Lamp1-RFP-expressing HEK293t cells, taken at different time points before (*t* = 0 s) and following (*t* = 1000 and 2000 s) the addition of DMSO (top row) or VPS34-IN1 and YM201636 (bottom row). Right: Average time course of

PI4KIIα-GFP fluorescence within Lamp1 signal in response to vehicle (DMSO, *n* = 14 cells) or VPS34-IN1 and YM201636 (*n* = 8) treatments. **e** Left: Confocal images of wild-type and ATG9A-depleted (sgATG9A) HEK293-Cas9 cells expressing PI4KIIα-GFP and treated with DMSO (control) or with VPS34-IN1 and YM201636. The bottom row shows enlarged views of PI4KIIα centered on endomembranes, within black rectangles. Right: Quantification of PI4KIIα-GFP fluorescence intensity within endomembrane regions relative to cytoplasm. DMSO: *n* = 9 cells; VPS34-IN1 + YM201636: *n* = 9; sgATG9A + VPS34-IN1 + YM201636: *n* = 12. **f** Same as (**e**), only in HEK293-Cas9 cells expressing P4M-YFP instead of PI4KIIα-GFP. DMSO: *n* = 16 cells; VPS34-IN1 + YM201636: *n* = 12; sgATG9A + VPS34-IN1 + YM201636: *n* = 15. All data are presented as mean ± s.e.m. For (**b**, **c**) statistical significance was determined using ordinary one-way ANOVA. For e and f, the statistical significance was determined using a two-way ANOVA.

To characterize the temporal dynamics of this process, we performed live-cell imaging of cells expressing PI4KIIα-GFP and Lamp1-RFP. Quantitative analysis revealed that PI4KIIα rapidly redistributed to lysosomal membranes within 15 min following acute application of VPS34-IN1 and YM201636, while no changes were observed in vehicle-treated controls (Fig. 3d). The kinetics of this response strongly suggest a direct effect on PI4KIIα trafficking rather than transcriptional regulation or secondary adaptations to PIKfyve complex dysfunction.

Recent studies have demonstrated that under starvation conditions, PI4KIIα and PI4KIIIβ co-traffic with the multispanning membrane protein ATG9A on vesicles originating from the TGN[37]. To determine whether a similar mechanism drives PI4KIIα redistribution following PIKfyve complex disruption, we co-expressed ATG9A-RFP and Lamp1-GFP in cells. Treatment with VPS34-IN1 and YM201636 significantly increased ATG9A accumulation on Lamp1-positive membranes compared to control conditions (Supplementary Fig. 3b). Critically, CRISPR-mediated knockdown of ATG9A in VPS34-IN1 and YM201636-treated cells effectively prevented the accumulation of both PI4KIIα and PI(4)P on lysosomal membranes, restoring their levels to those comparable with control conditions (Fig. 3e, f).

Together, these data establish that the PIKfyve complex regulates ATG9A-dependent trafficking of PI4KIIα from the TGN to lysosomes, providing a molecular mechanism for the observed PI(4)P redistribution following PIKfyve complex disruption.

## PIKfyve/Fig4/Vac14 deficiency drives ULK1-dependent PI4KIIα recruitment to lysosomes

ATG9A trafficking from the TGN is tightly regulated through ULK1/2-mediated phosphorylation[37–39]. Since ULK1 kinase activity is negatively regulated by mTOR signaling, and recent evidence has demonstrated that PIKfyve controls mTOR localization and activity at the lysosome[40], we hypothesized that PIKfyve complex disruption might enhance ULK1-dependent trafficking of ATG9A and PI4KIIα to lysosomes (Fig. 4a).

To test this hypothesis, we first examined mTOR localization in cells by immunofluorescence microscopy. Inhibition of either PIK3C3 with VPS34-IN1 or PIKfyve with YM201636 significantly reduced mTOR association with Lamp1-positive membranes (Fig. 4b), suggesting impaired lysosomal recruitment of mTOR. To directly measure the functional consequences of this altered localization, we employed two complementary approaches. First, we quantified the phosphorylation status of key mTORC1 substrates by western blot analysis. The ratio of phosphorylated-to-total ULK1 (at Ser757) and 4E-BP1 (at Thr37/46) both showed ~25% reductions following PIKfyve inhibition (Fig. 4c), consistent with decreased mTORC1 signaling output. Second, we utilized a bioluminescent resonance energy transfer (BRET)-based biosensor that specifically reports the phosphorylation status of ULK1 at Thr757, a direct mTORC1 target site[41]. In live-cell measurements, treatment with either VPS34-IN1 or YM201636 substantially decreased the BRET signal to levels comparable with those observed during treatment with the mTOR inhibitor Torin1 or expression of the phospho-deficient AIMTOR T757A mutant (Fig. 4d). Together, these orthogonal approaches confirm that PIKfyve inhibition attenuates mTORC1 kinase activity, particularly toward ULK1, a key regulator of vesicular trafficking.

Decreased ULK1 phosphorylation would be expected to release an inhibitory brake to influence ATG9A trafficking therefore we next investigated whether ULK1 activation drives enhanced trafficking of ATG9A and PI4KIIα to lysosomal membranes. Quantitative colocalization analysis revealed significantly increased ATG9A accumulation on Lamp1-positive structures following dual inhibition of PIK3C3 and PIKfyve (Fig. 4e). Critically, acute (2 h) treatment with the selective ULK1 inhibitor SBI-0206965[42] completely reversed this aberrant ATG9A distribution in PIKfyve-inhibited cells while having no effect on vehicle-treated controls, demonstrating the ULK1-dependency of this

process. Similarly, ULK1 inhibition normalized the elevated lysosomal recruitment of PI4KIIα observed in VPS34-IN1 and YM201636-treated cells (Fig. 4f).

To establish a direct causal link between ULK1 activation and lysosomal PI(4)P accumulation, we monitored the PI(4)P biosensor (P4M) in HEK293t cells and neurons treated with VPS34-IN1 and YM201636, with or without SBI-0206965. Quantification of P4M distribution on Lamp1-positive membranes demonstrated that ULK1 inhibition effectively prevented the PIKfyve inhibition-induced increases in lysosomal PI(4)P (Fig. 4g, Supplementary Fig. 3c). Despite reducing PI4KIIα and PI(4)P on lysosomal membranes, ULK1 inhibition did not rescue the vacuolation phenotype observed with PIKfyve complex disruption (Supplementary Fig. 3d). This suggests that lysosomal vacuolation involves multiple converging pathways downstream of PIKfyve disruption. Recent work has identified PDZD8 as a dedicated lysosomal vacuolator that responds to osmotic stress through PI(4)P-dependent recruitment and lipid transfer mechanisms[43]. Our findings indicate that while ULK1-dependent PI4KIIα trafficking contributes to the early stages of this response, the vacuolation process itself likely requires additional PDZD8-mediated mechanisms that operate on different timescales or through parallel signaling pathways independent of our 2 h ULK1 inhibition window.

These data establish a mechanistic signaling cascade wherein PIKfyve complex disruption reduces mTOR activity, thereby relieving inhibitory phosphorylation of ULK1 and promoting ULK1-dependent trafficking of ATG9A and PI4KIIα from the TGN to lysosomes. This pathway ultimately drives the accumulation of PI(4)P on lysosomal membranes, fundamentally altering lysosomal phosphoinositide composition.

## PIKfyve/Fig4/Vac14 loss of function promotes ULK1-mediated lysosomal membrane repair

Impaired lysosomal membrane integrity is a hallmark of several neurodegenerative diseases[44,45]. While the endosomal sorting complex required for transport (ESCRT) mediates a well-characterized membrane repair pathway[46,47], recent studies have identified phosphoinositide signaling as a crucial alternative mechanism for maintaining lysosomal membrane integrity[46,47]. This phosphoinositide-initiated membrane tethering and lipid transport (PITT) pathway is triggered by PI4KIIα recruitment to lysosomes, where it generates PI(4)P on the lysosomal membrane[48,49]. Lysosomal PI(4)P then establishes new membrane contact sites (MCS) between lysosomes and the endoplasmic reticulum (ER) by recruiting oxysterol-binding protein (OSBP) and related proteins, including ORP9. These lipid transfer proteins utilize the PI(4)P concentration gradient to counter-transport phosphatidylserine (PS) and cholesterol from the ER to the lysosome, strengthening membrane stability (Fig. 5a).

Given our findings that PIKfyve complex disruption triggers ULK1-dependent accumulation of PI4KIIα and PI(4)P on lysosomal membranes (Fig. 4f, g), we hypothesized that this pathway activates the PITT-mediated membrane repair system. To test this directly, we examined the localization and activity of OSBP following PIKfyve inhibition. Quantitative colocalization analysis revealed significant recruitment of OSBP to LAMP1-positive membranes in Apilimod-treated cells (Fig. 5b). We also indirectly assessed OSBP transfer activity using a functional assay where cells were treated with the OSBP-specific inhibitor OSW-1, which blocks OSBP-mediated counter-transport and causes PI(4)P accumulation at target membranes. Notably, PIKfyve-inhibited cells showed accelerated PI(4)P accumulation kinetics following OSW-1 treatment compared to controls (Fig. 5c), indicating enhanced OSBP activity at these sites. Since OSBP mediates counter-transport of cholesterol to membranes in exchange for PI(4)P, we examined cholesterol distribution using filipin staining. Quantitative imaging revealed significantly elevated cholesterol levels

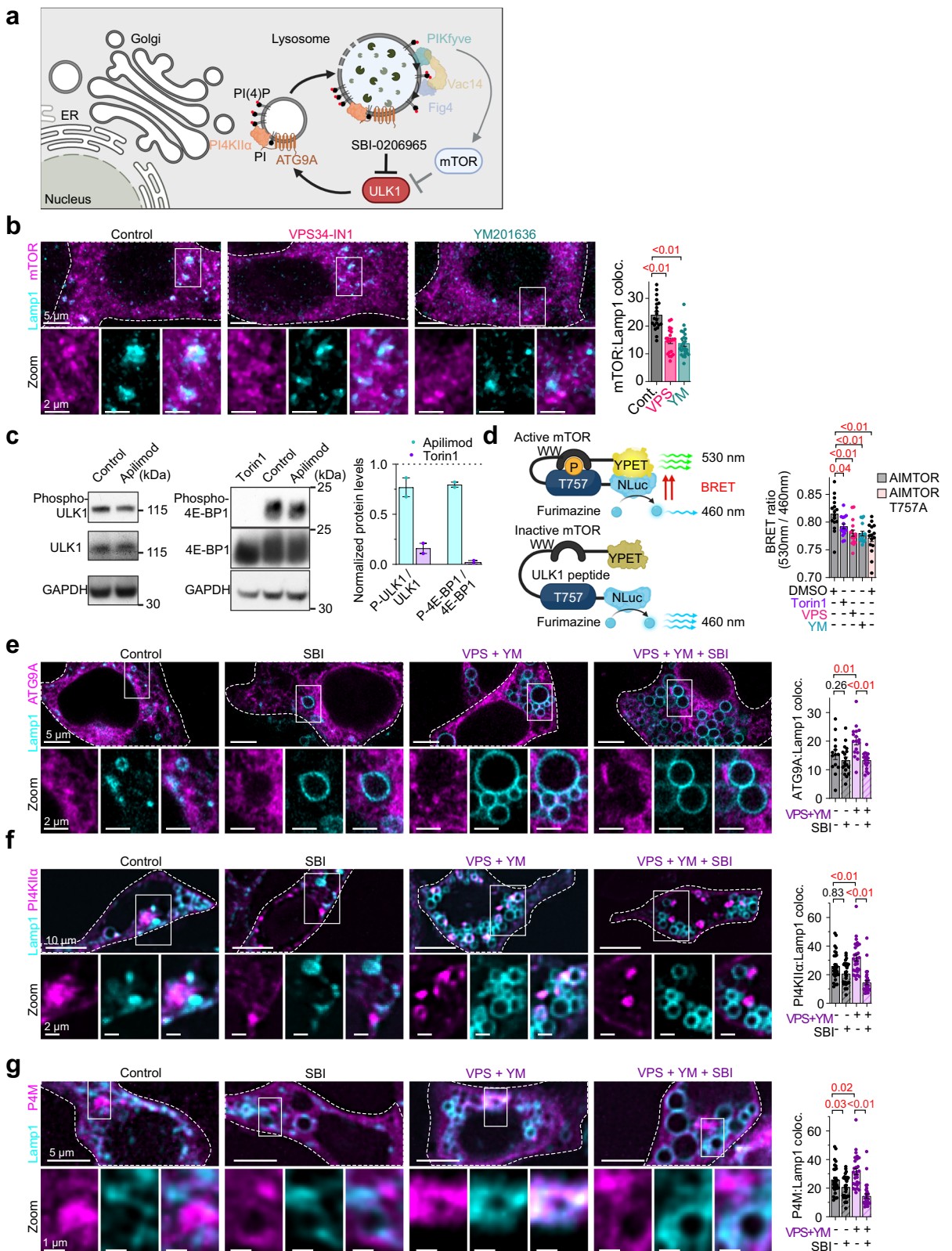

on intracellular membranes in PIKfyve-inhibited cells (Fig. 5d). In parallel, we investigated ORP9, which counter-transports phosphatidylserine (PS) for PI(4)P. Like OSBP, ORP9 showed robust recruitment to lysosomal membranes following PIKfyve inhibition (Fig. 5e), accompanied by significant accumulation of PS detected by the EVTx2 biosensor (Fig. 5f). Critically, co-treatment with the ULK1 inhibitor SBI-0206965 prevented ORP9 recruitment to lysosomal membranes and

abolished the accumulation of its cargo lipids, demonstrating that these events depend on ULK1-mediated signaling (Fig. 5d–f).

To directly test whether PIKfyve inhibition confers protection against lysosomal membrane damage, we challenged cells with L-leucyl-L-leucine methyl ester (LLOME), a lysosomotropic agent widely established to induce lysosomal membrane permeabilization. In control cells, LLOME exposure triggered robust recruitment of galectin-3

**Fig. 4 | PIKfyve complex deficiency drives ULK1-dependent PI4KIIα recruitment to lysosomes. a** Diagram of the working hypothesis: loss of PIKfyve complex upregulates ULK1-dependent ATG9A/PI4KIIα trafficking from TGN to promote lysosomal membrane repair. *Created in BioRender. Kutchukian, C. (2025)* https://BioRender.com/x628gxi. **b** Left: Confocal images of control and VPS34-IN1 or YM201636-treated HEK293t cells fixed and stained for mTOR (magenta) and Lamp1 (cyan). The bottom row shows magnified views of mTOR on lysosomal membranes. Right: Quantification of mTOR area within Lamp1 signal (%). Control: $n = 20$ cells; VPS34-IN1: $n = 21$; YM201636: $n = 25$. **c** Left: western blots of phospho-ULK1, total ULK1, phospho-4E-BP1, total 4E-BP1, and GAPDH from control, Apilimod- or torin1-treated cells. Right: quantification of protein bands from $n = 2$ independent experiments, normalized to GAPDH. **d** Left, schematic of the AIMTOR biosensor, *created in BioRender. Kutchukian, C. (2025)* https://BioRender.com/4czyzx8. Right: Quantification of the average 530 nm/460 nm ratio intensities in HEK297t cells expressing either AIMTOR or the non-responding mutant AIMTOR T757A, and treated with DMSO, VPS34-IN1, YM201636 or Torin1. For each condition, $n = 16$ fields of view. **e** Left: Confocal images of HEK293t cells expressing ATG9A-RFP (magenta) and Lamp1-GFP (cyan), and treated with either DMSO (control) or VPS34-

IN1 and YM201636, and pre-incubated with ULK-1 inhibitor SBI-0206965. The bottom row shows enlarged views of ATG9A centered on lysosomal regions. Right: Quantification of ATG9A area within Lamp1 signal (%). Control: $n = 14$ cells; SBI: $n = 18$; VPS34-IN1 + YM201636: $n = 17$; SBI + VPS34-IN1 + YM201636: $n = 18$. **f** Left: Confocal images of HEK293t cells expressing PI4KIIα-GFP (magenta) and Lamp1-RFP (cyan), and treated with either DMSO (control) or VPS34-IN1 and YM201636, and pre-incubated with SBI-0206965. The bottom row shows enlarged views of PI4KIIα centered on lysosomal regions. Right: Quantification of PI4KIIα area within Lamp1 signal (%). For each condition, $n = 14$ cells. **g** Left: Confocal images of HEK293t cells expressing P4M-mCherry (magenta) and Lamp1-GFP (cyan), and treated with either DMSO (control) or VPS34-IN1 and YM201636, and pre-incubated with SBI-0206965. The bottom row shows enlarged views of P4M in lysosomal regions. Right: Quantification of P4M area within Lamp1 signal (%). Control: $n = 30$ cells; SBI: $n = 30$; VPS34-IN1 + YM201636: $n = 28$; SBI + VPS34-IN1 + YM201636: $n = 30$. For (**b–g**) data are presented as mean ± s.e.m. For (**b**) the statistical significance was determined using ordinary one-way ANOVA; For (**d–g**) the statistical significance was determined using a two-way ANOVA.

---

to lysosomal membranes, a hallmark indicator of membrane damage and permeability (Fig. 5g). Remarkably, pre-treatment with Apilimod reduced LLOME-induced galectin-3 recruitment to lysosomes. This protective effect was abolished by ULK1 inhibition, demonstrating its dependence on the ULK1-PI4KIIα-PI(4)P pathway. Finally, we evaluated lysosomal function in HEK293t cells expressing LAMP1-pHluorin, a lysosome-targeted, pH-sensitive GFP variant whose fluorescence is quenched in highly protonated environments. In control cells, exposure to an alkaline solution ($NH_4Cl$, pH 8) abolished the lysosomal proton gradient, as evidenced by a rapid increase in pHluorin fluorescence intensity (Fig. 5h). In contrast, cells pre-treated with LLOME exhibited a higher baseline fluorescence upon $NH_4Cl$ exposure, indicating reduced lysosomal acidity. Notably, Apilimod-pretreated cells maintained lysosomal acidification following LLOME challenge, consistent with preserved membrane integrity. Together, these findings demonstrate that PIKfyve complex disruption not only increases lysosomal PI(4)P levels but also functionally enhances lysosomal membrane integrity through coordinated recruitment of lipid transfer proteins and enrichment of membrane-stabilizing lipids, revealing an adaptive mechanism for cytoprotection against lysosomal stress.

## PIKfyve/Fig4/Vac14 complex controls mitochondrial morphology and function

Crosstalk between mitochondria and lysosomes is crucial to neuronal health, as emphasized by the close relationship between mitochondrial defects and neurodegeneration in various lysosomal storage disorders. Considering our findings that PIKfyve disruption drives the redistribution of PI(4)P to lysosomes and that recent work has implicated PI(4)P in regulating mitochondrial fission[50,51], we investigated whether these changes impact mitochondrial homeostasis (Fig. 6a).

First, we asked if the cellular organization of ER, lysosomes, and mitochondria was altered following Apilimod treatment. Consistent with Apilimod facilitating enhanced 3-way contacts between the triad of organelles, transmission electron microscopy (Fig. 6b) and super-resolution confocal microscopy (Fig. 6c) revealed that PIKfyve inhibition significantly increased the frequency of ER-lysosome-mitochondria three-way contacts, with contact sites increasing ~2.5-fold following Apilimod treatment compared to controls (Fig. 6c). These contacts were specifically enriched with ORP1L, which accumulated at the interface between lysosomes and mitochondria following PIKfyve inhibition (Fig. 6d). This recruitment was strictly ULK1-dependent, as co-treatment with SBI-0206965 completely prevented ORP1L accumulation at these contact sites, as well as their occurrence, establishing a direct link between our identified ULK1-PI4KIIα-PI(4)P pathway and organellar contact formation.

At these three-way junction sites, immunofluorescence analysis revealed substantial enrichment of the mitochondrial fission protein Drp1, with quantitative measurements showing a significant increase in Drp1-positive contacts following PIKfyve inhibition (Fig. 6e). Consistent with enhanced fission machinery recruitment, confocal microscopy analysis revealed extensive mitochondrial fragmentation in cells treated with VPS34-IN1 and YM201636 (Fig. 7a). Importantly, this fragmentation was completely reversed by co-treatment with the ULK1 inhibitor SBI-0206965, demonstrating that altered mitochondrial dynamics directly result from ULK1 activation (Fig. 7b). Similar trends were observed in Fig4$^{-/-}$ and Vac14$^{-/-}$ cells (Supplementary Fig. 4a).

To establish whether lysosomal PI(4)P directly mediates these effects, we employed two complementary approaches. First, treatment with the selective PI4KIIα inhibitor PI-273 effectively reduced PI(4)P levels (Supplementary Fig. 4b) and restored normal mitochondrial morphology in PIKfyve-inhibited cells, with quantitative analysis confirming increased mitochondrial length and branching compared to YM201636 treatment alone (Fig. 7c). Second, we expressed a mitochondrially-targeted PI(4)P phosphatase (Sac1-Fis1[52]) to locally reduce PI(4)P accumulation on mitochondria. Expression of Sac1-Fis1 in PIKfyve-inhibited cells effectively prevented mitochondrial fragmentation compared to cells expressing a catalytically inactive Sac1 (Sac1$^{C389S}$-Fis1; Fig. 7d). These orthogonal approaches provide evidence that local mitochondrial PI(4)P accumulation, potentially from lysosomes via ORP1L at three-way contacts, directly drives the observed changes in mitochondrial morphology.

To assess whether PIKfyve-dependent mitochondrial fragmentation affects cellular bioenergetics, we performed comprehensive real-time oxygen consumption rate (OCR) measurements. PIKfyve inhibition significantly enhanced all measured parameters of mitochondrial respiration, including basal respiration, maximal respiratory capacity, proton leak, and ATP-coupled respiration (Fig. 7e). This enhanced bioenergetic state was ULK1-dependent, as co-treatment with SBI-0206965 normalized most OCR parameters to control levels. While mitochondrial fragmentation often correlates with dysfunction in pathological contexts, our findings are consistent with emerging evidence that adaptive fragmentation can enhance respiratory efficiency. This occurs through increased surface area-to-volume ratios that facilitate substrate exchange and optimize electron transport chain function, as demonstrated in brown adipocyte thermogenesis[53], cardiac exercise adaptation[54], and immune cell activation[55]. These findings demonstrate that PIKfyve inhibition triggers a coordinated cellular response that enhances both mitochondrial morphological dynamics and metabolic capacity.

Collectively, our findings suggest a pathway in which PIKfyve complex dysfunction promotes ULK1-dependent recruitment of

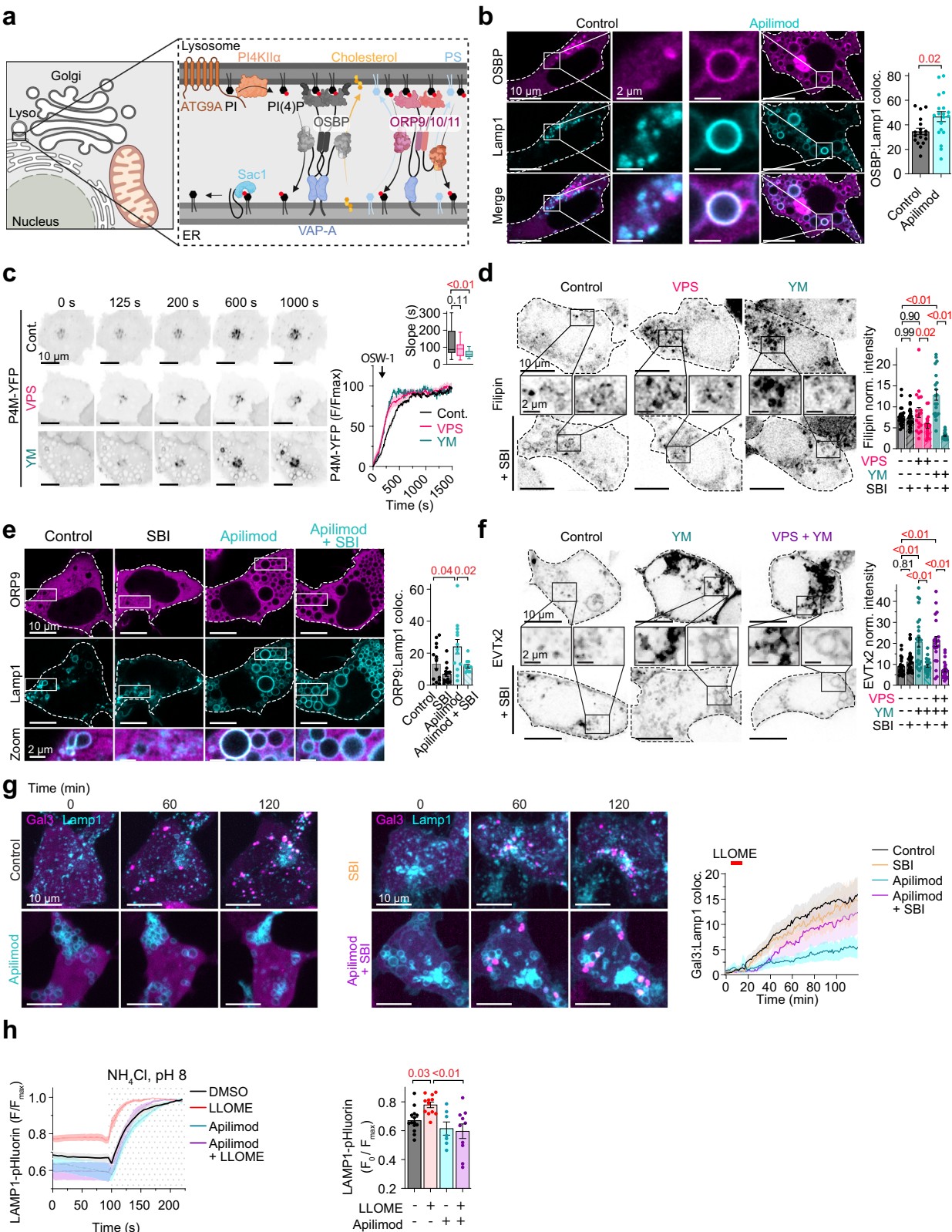

PI4KIIα to lysosomes, leading to increased lysosomal PI(4)P levels. This is accompanied by the formation of closer contact sites between lysosomes, ER, and mitochondria, enriched in ORP1L and Drp1. These contacts may provide a platform for PI(4)P transfer to mitochondria and promote mitochondrial fission through Drp1 recruitment. Consistently, PIKfyve dysfunction enhances mitochondrial fission in a ULK1-dependent manner and is associated with increased respiratory capacity. This pathway represents a short-term adaptive mitochondrial response that may help maintain energy homeostasis under conditions of cellular stress associated with lysosomal dysfunction.

## Discussion

In this study, we have identified key components of a molecular pathway linking the PIKfyve complex to lysosomal membrane repair

**Fig. 5 | PIKfyve complex inhibition promotes lysosomal membrane repair.**
**a** Diagram illustrating the phosphoinositide pathway leading to lysosomal membrane repair. *Created in BioRender. Kutchukian, C. (2025)* https://BioRender.com/hl99foh. **b** Left: Confocal images of DMSO (control) or Apilimod-treated HEK293t cells and expressing GFP-OSBP (top), Lamp1-RFP (middle), and merge (bottom). Insets show enlarged views of OSBP on lysosomes. Right: Quantification of average OSBP fluorescence intensity on Lamp1 membranes. Control: $n = 18$ cells; Apilimod: $n = 19$. **c** Left: representative live AiryScan time series images from COS7 cells expressing P4M-YFP under control, VPS34-IN1, and YM201636 treated conditions. Right: Quantification of normalized P4M-YFP intensity (F/Fmax) following treatment with 20 nM OSW-1. Inset: quantification of slope kinetics following OSW-1 treatment. Control: $n = 34$ cells; VPS34-IN1: $n = 21$; and YM201636: $n = 24$. **d** Left: Confocal images of DMSO (control) VPS34-IN1 or YM201636-treated HEK293t cells, pre-incubated with SBI-0206965 and fixed and stained for filipin. Insets show enlarged views of endomembrane regions. Right: Quantification of average filipin fluorescence intensity relative to the cytoplasm. Control: $n = 21$ cells; SBI: $n = 24$; VPS34-IN1: $n = 18$; SBI + VPS34-IN1: $n = 20$; YM201636: $n = 20$; SBI + YM201636: $n = 20$. **e** Left: Confocal images of DMSO (control) or Apilimod-treated HEK293t cells, pre-incubated with SBI-0206965 and expressing ORP9-GFP (top) and Lamp1-RFP (middle). Bottom row shows enlarged views of ORP9 on lysosomes. Right: Quantification of average ORP9 fluorescence intensity on Lamp1 membranes. Control: $n = 14$ cells; SBI: $n = 13$; Apilimod: $n = 13$; Apilimod + SBI: $n = 12$. **f** Left: Confocal images of DMSO (control), VPS34-IN1 and/or YM201636-treated HEK293t cells, pre-incubated with SBI-0206965 and expressing EGFP-EVTx2. Insets show enlarged views of endomembrane regions. Right: Quantification of average EVTx2 fluorescence intensity relative to the cytoplasm. Control: $n = 29$ cells; SBI: $n = 26$; YM201636: $n = 26$; SBI + YM201636: $n = 24$; VPS34-IN1 + YM201636: $n = 21$; SBI + VPS34-IN1 + YM201636: $n = 26$. **g** Left: confocal, time series micrographs of HEK293t cells expressing Gal3-mCherry (magenta) and Lamp1-GFP (cyan) and treated with DMSO (control) or Apilimod, with or without SBI-0206965 co-treatment. Right: quantification analysis of Gal3 signal on Lamp1 membranes. Control: $n = 8$ cells; SBI: $n = 9$; Apilimod: $n = 8$; Apilimod + SBI: $n = 8$. **h** Left: time series analysis from HEK293t cells expressing LAMP1-pHluorin and treated with DMSO, LLOME, Apilimod, or LLOME and Apilimod prior to stimulation with $NH_4Cl$, pH 8. Right: quantification of changes in LAMP1-pHluorin intensity. For (**b**–**f** and **h**) data are presented as mean ± s.e.m. For (**b**) a two-tailed $t$-test was used to determine statistical significance while for (**c** and **h**), a one-way ANOVA was used, and for (**d**–**f**) the statistical significance was determined using a two-way ANOVA.

---

and mitochondrial bioenergetics. We show that loss of PIKfyve complex integrity or pharmacological inhibition of PIKfyve triggers ULK1-dependent trafficking of ATG9A and PI4KIIα from the TGN to lysosomes, leading to elevated PI(4)P levels on lysosomal membranes. This increase facilitates cholesterol and phosphatidylserine transport to lysosomal membranes, promoting membrane repair. Concurrently, we observe increased mitochondrial fragmentation and enhanced respiratory capacity. We propose a model wherein the PIKfyve complex acts as a regulatory brake on lysosomal membrane repair and mitochondrial dynamics under basal conditions (Fig. 8). When this complex is disrupted, as occurs in certain neurodegenerative conditions, or when $PI(3,5)P_2$ levels are decreased, this inhibition is lifted, potentially representing a compensatory mechanism to preserve cellular function under stress.

On lysosomal membranes, PIKfyve, Fig4, and Vac14 form a ternary complex that tightly regulates $PI(3,5)P_2$ synthesis and turnover[10,11,56]. Recent structural biochemistry studies have revealed that Vac14 oligomers act as a scaffold, positioning PIKfyve near Fig4, enabling Fig4 to regulate PIKfyve kinase activity through phosphorylation[12,57]. Paradoxically, despite Fig4's inherent phosphatase activity, loss-of-function mutations in either Fig4 or Vac14 lead to decreased $PI(3,5)P_2$ levels[13,14], underscoring the complex interplay within this ternary complex and highlighting the importance of its structural integrity.

Our data strongly suggest that the absence of $PI(3,5)P_2$, rather than PI(3)P accumulation or physical disruption of the complex, drives the cellular effects we observe. At lysosomal membranes, $PI(3,5)P_2$ serves as a critical anchoring lipid essential for mTOR recruitment and activation. Disruption of the PIKfyve/Fig4/Vac14 complex reduces $PI(3,5)P_2$ levels, resulting in mTOR dissociation from lysosomal membranes and a subsequent attenuation of its kinase activity[58,59]. This small decrease in mTOR activity relieves inhibitory phosphorylation of ULK1[60], enabling ULK1 to orchestrate ATG9A-dependent trafficking of PI4KIIα to lysosomes where it catalyzes PI(4)P synthesis. However, the modest population-averaged decreases in canonical mTORC1 readouts (~25% reduction in ULK1 and 4E-BP1 phosphorylation) contrast sharply with the robust ULK1-dependent functional phenotypes we observe. This apparent disconnect suggests that ULK1 regulation under PIKfyve inhibition involves additional complexity beyond bulk mTORC1 suppression. Several mechanisms may contribute including ultrasensitive threshold effects characteristic of autophagy signaling, localized mTORC1 inhibition that exceeds population averages, or direct PIKfyve-ULK1 interactions that influence ULK1 complex stability[61]. Defining these regulatory mechanisms represents an important avenue for future investigation to fully understand how modest biochemical perturbations translate into pronounced cellular adaptations.

The shift in signature lysosome phosphoinositide from $PI(3,5)P_2$ to PI(4)P represents a conserved lipid-based signaling switch fundamental to lysosomal adaptation across diverse cellular contexts[62]. Similar lysosomal PI(4)P elevations occur during nutrient deprivation to coordinate proteolysis and mTORC1 repression, suggesting that precisely regulated $PI(3)P/PI(3,5)P_2$ ratios establish critical homeostatic setpoints. When perturbed, these trigger compensatory cascades using PI(4)P as a lysosomal beacon to coordinate proteolysis, growth suppression, and membrane repair. Our findings likely represent integrated cellular responses involving multiple converging mechanisms. PI4KIIα redistribution involves both ULK1-dependent trafficking and altered retrograde transport, as PIKfyve normally regulates endosome-to-TGN trafficking[63]. Similarly, the phenotypes we observe reflect both direct signaling events and secondary adaptations to phosphoinositide perturbations, highlighting the complex nature of cellular homeostatic networks. This mechanistic framework may explain the paradoxical therapeutic effects of acute PIKfyve inhibition in neurodegenerative disease models, despite the pathological consequences of genetic PIKfyve/Fig4/Vac14 dysfunction. Short-term activation of these compensatory pathways, enhancing lysosomal repair, mitochondrial function, and autophagic clearance, could provide transient protective effects under specific disease conditions. However, chronic dysregulation of phosphoinositide metabolism, as occurs in genetic disorders, likely overwhelms these adaptive mechanisms and proves ultimately detrimental.

Linking PIKfyve to elevations in lysosomal PI(4)P is the ULK1-dependent trafficking of PI4KIIα from the TGN to lysosomes. Mammalian cells express four distinct PI 4-kinase isoforms, with PI4KIIα and PI4KIIIβ primarily localized to the TGN network, where they generate PI(4)P from PI. Our work demonstrates that disruption of the PIKfyve complex causes PI4KIIα, but not PI4KIIIβ, to redistribute from the TGN to lysosomal membranes. There, PI4KIIα increases PI(4)P production, which correlates with elevated cholesterol and PS levels. These findings align with recently identified lysosomal membrane repair mechanisms where PI4KIIα-mediated PI(4)P synthesis recruits OSBP and ORPs to ER-lysosome contact sites, facilitating the transfer of cholesterol and PS in exchange for PI(4)P. The interplay between this PI(4)P-dependent pathway of membrane repair and other lysosomal repair mechanisms, such as the ESCRT machinery, remains an open area for future research. This relationship could provide crucial insights into the cell's comprehensive response to lysosomal damage under various stress conditions.

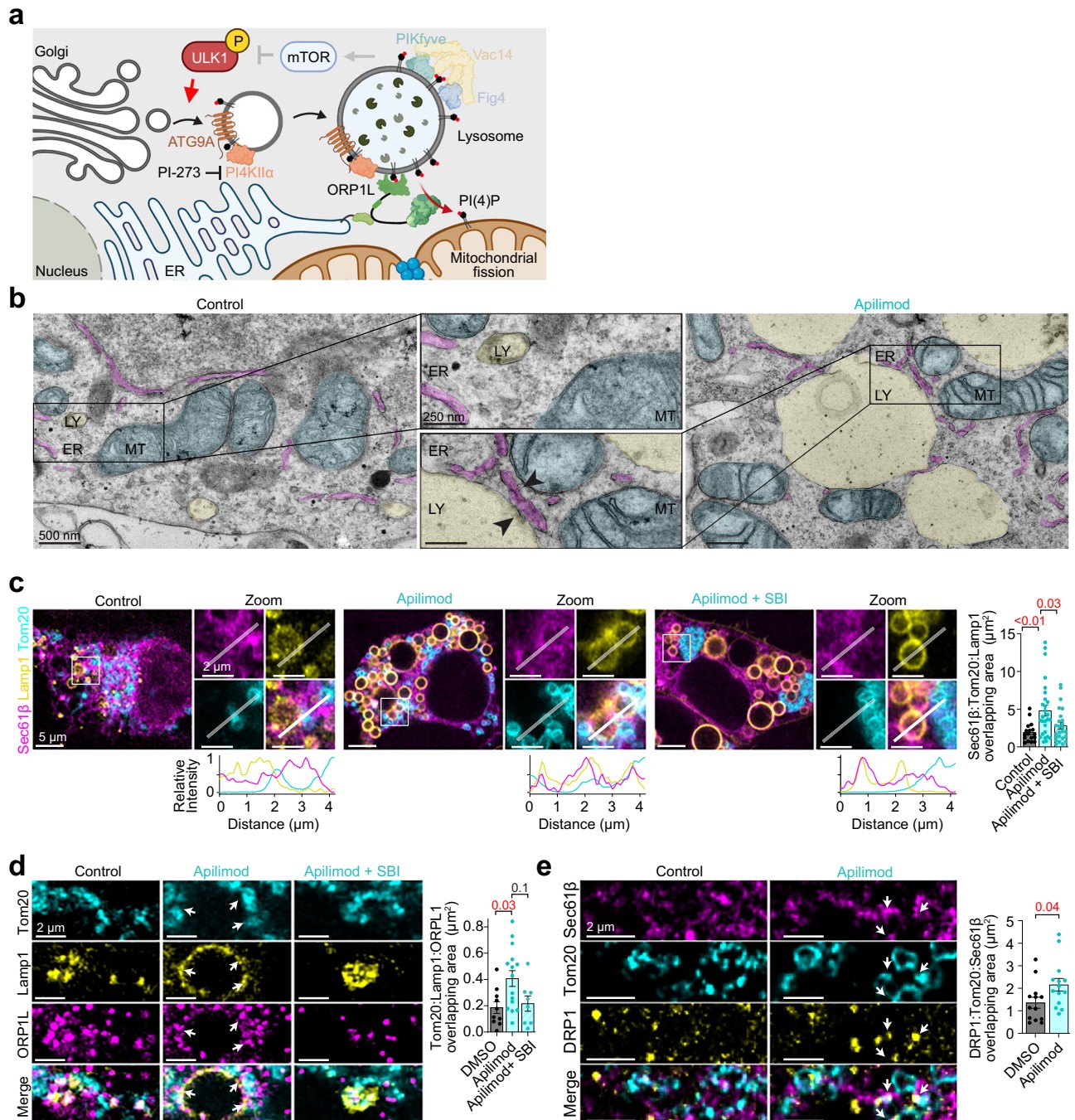

**Fig. 6 | PIKfyve complex inhibition increases ER–Lysosome–Mitochondrial 3-way membrane contact sites. a** Diagram of the working hypothesis: PIKfyve complex loss of function leads to mitochondrial fragmentation through increased ORP1L-mediated transfer of lysosomal PI(4)P at ER–lysosome–mitochondrial contacts. *Created in BioRender. Kutchukian, C. (2025)* https://BioRender.com/p04pmls. **b** Representative transmission electron micrographs from DMSO-treated (control) or Apilimod-treated HEK293t cells. Zoomed areas demonstrate the organization of ER (magenta), lysosome (yellow), and mitochondria (cyan). The images are representative of *n* = 2 independent biological experiments. **c** Left: representative super-resolution confocal images from HEK293t cells expressing Sec61β (ER marker, magenta), Lamp1 (lysosome marker, yellow), and Tom20 (mitochondrial marker, cyan) in control (left), Apilimod (middle), or Apilimod and SBI-0206965 (right)

treated cells. White boxes represent zoomed areas. Intensity line scans are taken from the white line in the zoomed areas. Right: quantification of signal overlap. Control: *n* = 17 cells; Apilimod: *n* = 27; Apilimod + SBI: *n* = 25. **d** Left: Representative confocal images from HEK293t cells expressing Lamp1 (yellow) and fixed and stained for Tom20 (cyan) and ORP1L (magenta) in control (left), Apilimod (middle), or Apilimod and SBI-0206965 (right) treated cells. Right: quantification of signal overlap. Control: *n* = 10 cells; Apilimod: *n* = 15; Apilimod + SBI: *n* = 8. **e** Same as (**c**) only in control and Apilimod-treated cells expressing Sec61β (magenta) and immunolabeled for Drp1 (yellow) and Tom20 (cyan). Control: *n* = 13 cells; Apilimod: *n* = 14. For (**c**–**e**) data are presented as mean ± s.e.m. For **c** and **d**, a one-way ANOVA was used, for (**e**) a two-tailed student's *t*-test was used to determine statistical significance.

A particularly unique aspect of our findings is the identification of PI(4)P-dependent three-way contacts between the ER, lysosomes, and mitochondria that emerge following PIKfyve complex inhibition. These specialized membrane junction sites serve as critical

communication hubs that coordinate the adaptive response across multiple organelles. At these three-way junction sites, ORP1L accumulates in a ULK1-dependent manner, where it appears to facilitate PI(4)P availability at mitochondrial membranes. While our data

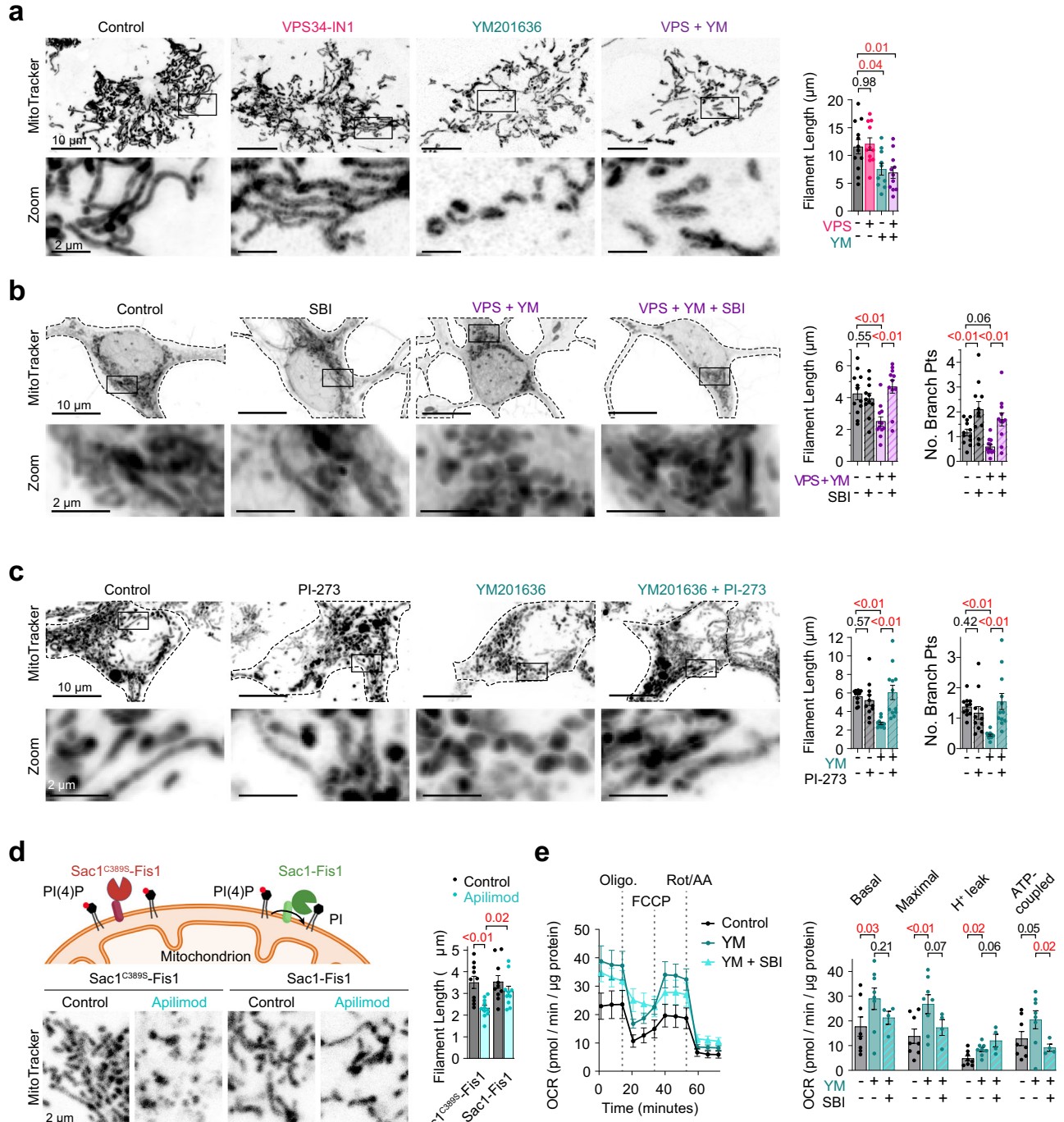

**Fig. 7 | PIKfyve complex inhibition influences mitochondrial morphology and function. a** Left: Super-resolution confocal images of DMSO (control) and VPS34-IN1 and/or YM201636-treated COS7 stained with MitoTracker Deep Red. Right: Quantification of the average mitochondrial filament length (μm). Control: n = 12 cells; VPS34-IN1: n = 11; YM201636: n = 10; dual treatment: n = 11. **b** Left: Confocal images of DMSO (control) and VPS34-IN1 and YM201636-treated live cortical neurons stained with MitoTracker Deep Red, and incubated with or without SBI-0206965. Right: Quantification of the average mitochondrial filament length (μm, left graph) and the average number of branching points per filament (right graph). Control: n = 12 neurons; SBI: n = 11; VPS34-IN1 + YM201636: n = 12; SBI + VPS34-IN1 + YM201636: n = 11. **c**, Same as (**b**), only in DMSO (control) and YM201636-treated HEK293t cells pre-incubated with or without the PI4KIIα inhibitor PI-273 (n = 11 cells for each condition). **d** Top: schematic of different Sac1 plasmids.

Created in BioRender. Kutchukian, C. (2025) https://BioRender.com/7fms4aj. Bottom: Confocal images of HEK293t cells stained for MitoTracker Deep Red and expressing the non-catalytic mutant Sac1$^{C389S}$-Fis1 (left) or Sac1-Fis1 (right) and treated with/without DMSO (control) or Apilimod for 2 hr. Right: analysis of mitochondrial filament length. For Sac1$^{C389S}$-Fis1: n = 10 cells (control) and 11 (Apilimod); For Sac1-Fis1: n = 10 cells for each condition. **e** Left: Time course of oxygen consumption rate (OCR) in control (n = 8 technical replicates) and YM201636-treated HEK293t cells pre-incubated with (n = 4) or without SBI-0206965 (n = 8), upon the sequential addition of oligomycin, FCCP and Rotenone/Antimycin A. Right: Quantification of basal, maximal, proton leak-coupled and ATP production-coupled OCR, normalized to the protein content. For (**a–e**) data are presented as mean ± s.e.m. For (**a**), statistical significance was determined using a one-way ANOVA, while for (**b–e**) a two-way ANOVA was used.

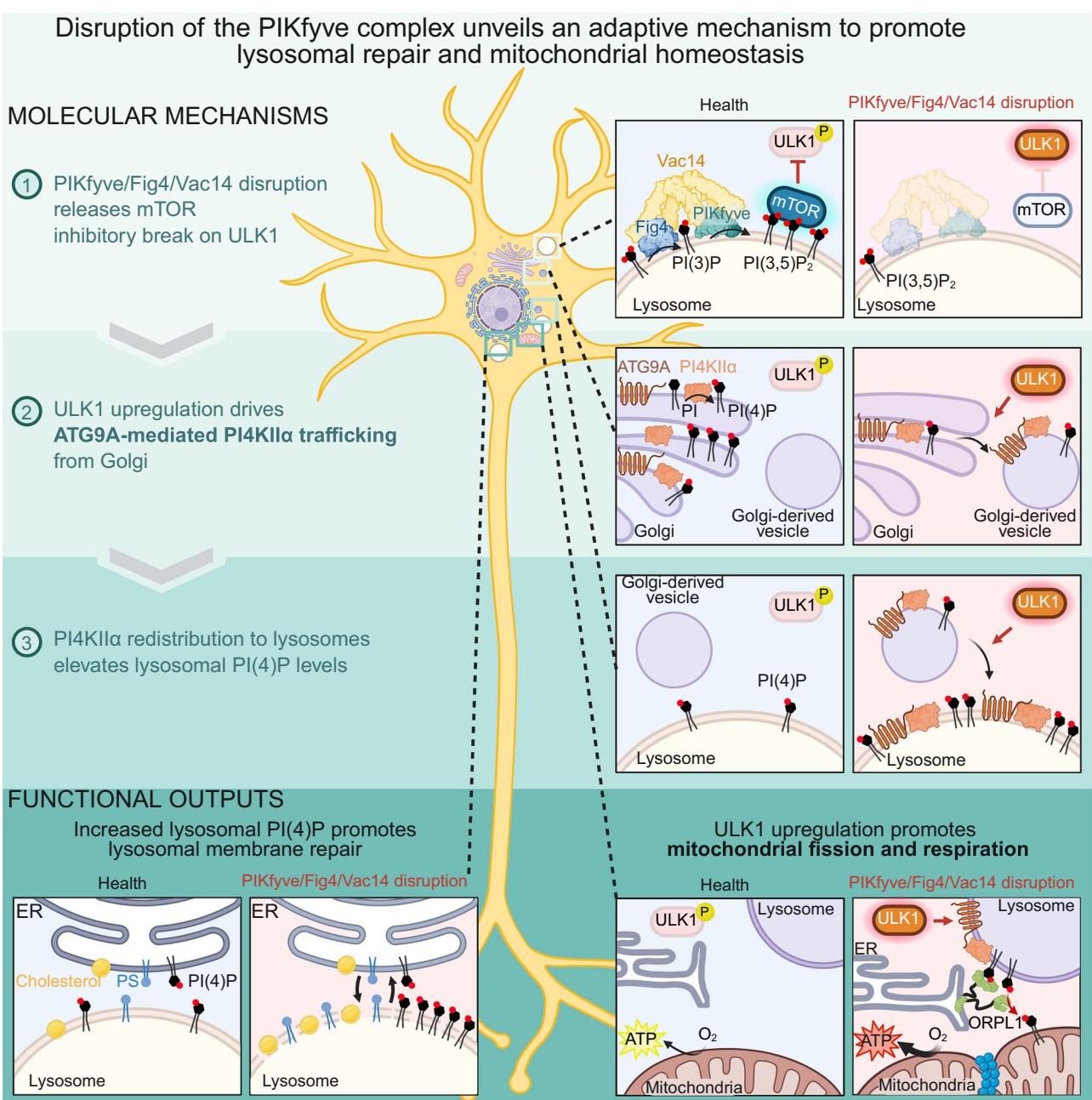

**Fig. 8 | Model: PIKfyve complex regulates lysosomal membrane repair and mitochondrial homeostasis.** *Created in BioRender. Kutchukian, C. (2025)* https://BioRender.com/w9fjsyq.

demonstrate that both lysosomal PI(4)P elevation and ORP1L recruitment are necessary for the observed mitochondrial effects, the precise mechanism, whether through direct lipid transfer, local scaffolding, or membrane tethering, remains to be definitively established.

These three-way contacts represent specialized microdomains where the ER, which contains ORP1Ls interacting partner, VAP-A, can coordinate with lysosomes that have accumulated PI(4)P following PIKfyve inhibition, to simultaneously transfer PI(4)P lipids to mitochondria. The enrichment of mitochondrial fission protein Drp1 at these contact sites provides a direct mechanistic link between lysosomal PI(4)P elevation and the observed mitochondrial fragmentation[50,64,65]. Increased fission enables quality control by segregating and eliminating damaged mitochondrial components, improving the overall efficiency of the mitochondrial network. The resulting smaller mitochondria have a higher surface area-to-volume ratio, facilitating greater substrate uptake for ATP production. Our

experimental manipulations provide mechanistic evidence for a direct role of lysosomal PI(4)P in orchestrating mitochondrial dynamics through specialized membrane contact sites. The targeted depletion of PI(4)P from mitochondria using a mitochondrially-targeted Sac1 phosphatase prevents fragmentation despite persistent lysosomal PI(4)P elevation, establishing that mitochondrial PI(4)P specifically, rather than simply elevated lysosomal PI(4)P, drives the observed morphological changes. Similarly, ULK1 inhibition rescues mitochondrial fragmentation while simultaneously preventing ORP1L recruitment to three-way contacts, connecting these molecular events in a clear signaling cascade. These complementary approaches demonstrate that PI(4)P functions as a lipid messenger that can be transferred between organelles at these specialized junction sites to precisely regulate mitochondrial form and function. The spatial specificity of this transfer, likely occurring at discrete three-way contacts between ER, lysosomes, and mitochondria, enables cells to coordinate adaptive

responses across multiple organelles simultaneously. This inter-organellar lipid signaling network represents a previously unappreciated mechanism by which cells integrate lysosomal stress responses with mitochondrial adaptations, potentially explaining how seemingly distinct organellar pathologies often co-occur in neurodegenerative and lysosomal storage disorders.

PIKfyve inhibition not only enhances lysosomal membrane repair and mitochondrial energetics but also triggers TFEB nuclear translocation[40], a master regulator of autophagy and lysosomal biogenesis, and lysosomal vacuolation, which may be helpful to prevent lysosomal rupture[43]. These beneficial cellular responses position PIKfyve as a potential therapeutic target for disorders characterized by lysosomal dysfunction, particularly neurodegenerative diseases where failure of lysosomes to clear protein aggregates is a key feature.

Recent studies provide compelling evidence supporting the therapeutic potential of PIKfyve inhibition across multiple neurodegenerative models. In cellular models of Parkinson's disease, PIKfyve inhibition reduced α-synuclein spread from lysosomes into the cytosol[66]. In Alzheimer's disease models, it mitigated tau aggregation in cultured neurons[67], and in tau-V337M cortical organoids modeling frontotemporal dementia[68], it rescued neuronal cell death. Furthermore, in cardiomyocytes, loss of PIKfyve function prevents myocardial apoptosis and hypertrophy by boosting mitochondrial function[69]. These findings suggest that PIKfyve inhibition could offer a multifaceted approach to addressing lysosomal dysfunction by simultaneously enhancing membrane repair mechanisms and promoting lysosomal biogenesis. However, the sustainability of chronic PIKfyve inhibition remains an ongoing debate. While acute inhibition shows promise, the potential side effects and compensatory mechanisms that might arise from prolonged PIKfyve suppression need thorough investigation. Beyond the neurodegeneration associated with FIG4 and VAC14 loss-of-function mutations[13,15–18], PIKfyve downregulation has been described in prion-infected neurons[70] and likely accounts for spongiosis and lysosomal defects leading to neurotoxicity. Future challenges will be to determine the optimal balance in modulating PIKfyve activity to leverage the protective effects while avoiding detrimental consequences associated with prolonged disruption of PI(3,5)$P_2$ homeostasis.

While our findings provide substantial evidence for the proposed pathway, several limitations should be acknowledged. First, our lipidomics approach cannot distinguish between PIP regioisomers, limiting precise quantification of individual phosphoinositide species. Second, the modest population-averaged effects on canonical mTORC1 substrates relative to the robust ULK1-dependent phenotypes suggest additional regulatory mechanisms that require further elucidation. Third, while our data strongly support ORP1L involvement in facilitating mitochondrial PI(4)P availability, direct demonstration of lipid transfer mechanisms awaits development of appropriate methodological approaches. Finally, the temporal relationships and causal interdependencies among the Golgi remodeling events we observe require more detailed kinetic analysis. These limitations highlight important avenues for future research that will deepen our mechanistic understanding of this adaptive cellular response.

Our study identifies key components of an adaptive cellular response linking PIKfyve complex dysfunction to lysosomal membrane repair and mitochondrial bioenergetics through ULK1-dependent trafficking of PI4KIIα. While important mechanistic questions remain, these findings provide a foundation for understanding how cells coordinate organellar responses to phosphoinositide perturbations. This pathway, triggered by the loss of PI(3,5)$P_2$, appears to be a compensatory mechanism that stabilizes lysosomal membranes and enhances mitochondrial function in response to cellular stress. The identification of three-way contacts between the ER, lysosomes, and mitochondria provides a critical missing link in understanding how these organelles coordinate their activities during cellular adaptation to stress. While these findings suggest potential therapeutic avenues for neurodegenerative disorders and possibly cancer, the long-term effects of modulating this pathway require further investigation. Future research should focus on optimizing the balance between leveraging the protective effects of this mechanism and avoiding the detrimental consequences of prolonged PI(3,5)$P_2$ homeostasis disruption, potentially leading to more targeted and effective treatments for a range of pathological conditions.

## Methods

### DNA plasmids, antibodies and chemicals
See Supplementary Table 1.

### Cell culture, transfection and pharmacological treatments
All cell lines were cultured in a 5% $CO_2$ incubator at 37 °C. Control, *FIG4* knock-out (Fig4$^{-/-}$), and *VAC14* knock-out (Vac14$^{-/-}$) HAP1 cells were kindly provided by Dr. J. O. Kitzman[23] (University of Michigan, Ann Arbor, MI) and were grown in Iscove's Modified Dulbecco's Medium (IMDM) supplemented with 10% fetal bovine serum (FBS) and 1% penicillin/streptomycin. HEK293t (Sigma, Cat #96121229), HEK293 stably expressing Cas9 (HEK293 Cas9, ATCC, Cat #CRL-1573Cas9), and COS7 cells (kindly gifted from Dr. J. Nunnari, University of California, Davis, CA) were grown in Dulbecco's Modified Eagle Medium (DMEM) supplemented with 10% FBS and 0.2% penicillin/streptomycin. Cells intended for confocal imaging were seeded onto 25 mm diameter #1.5 cover glasses, and plasmid DNA transfections were performed 24 h before experiments using either Lipofectamine 2000 (Invitrogen) for cortical neurons and HEK293t cells, or Lipofectamine LTX (Invitrogen) and jetOPTIMUS (Polypus) for HAP1 and COS7 cells. For genetic knock-down experiments, HEK293 Cas9 cells were co-transfected for 48 h with 50 nM single-guide RNA targeting ATG9A along with either P4M-YFP or PI4KIIα-GFP DNA plasmids, using Lipofectamine LTX reagent. For PIKfyve and PIK3C3 pharmacological inhibition, cells were incubated for 2 h at room temperature with 500 nM Apilimod or overnight with 1 μM YM201636 and/or 1 μM VPS34-IN1, respectively. SBI-0206965, PI-273, and Torin1 treatments were added to the cell culture medium 2 h before imaging, at 1 μM, 2 μM and 250 nM, respectively.

### Lipid mass spectrometry
Cell harvest and lysis: HAP1 wild-type (WT), Fig4$^{-/-}$, and Vac14$^{-/-}$ cells were harvested on ice into RIPA buffer (Thermo Scientific, Cat. 89900) supplemented with cOmplete Mini, EDTA-free protease inhibitor cocktail (Roche, Cat. 11836170001) and incubated for 15 min at 4 °C. Lysates were sonicated in a chilled bath sonicator at 4 °C and clarified by centrifugation at 13,600 × g for 20 min at 4 °C to isolate the post-nuclear supernatant.

Lipid extraction and methylation: Endogenous lipids and internal standards were extracted using sequential n-butanol and chloroform partitioning. Extraction efficiency was assessed with 37:4 PIP synthetic standards added prior to extraction. Briefly, 100 μl n-butanol was added to each clarified lysate, vortexed vigorously, and incubated on ice for 10 min. Next, 15 μl 6 N HCl was added, the mixture was vortexed, incubated on ice for 10 min, and centrifuged at 26,300 × g for 2.5 min at 4 °C (Thermo Fisher Sorvall microcentrifuge). The upper n-butanol phase (85 μl) was removed and retained. The remaining aqueous phase was re-extracted four times using the same procedure: once with 100 μl n-butanol and three times with 100 μl chloroform. The two n-butanol (upper) extracts and three chloroform (lower) extracts were combined, centrifuged for 2.5 min, and taken to dryness under $N_2$ using a Biotage TurboVap prior to derivatization. For methyl derivatization of phosphate hydroxyls, dried lipid extracts (or standards alone) were resuspended in 90 μl methanol/dichloromethane

(4:5, v/v), vortexed, and reacted with 20 μl 2 M trimethylsilyl-diazomethane (TMS-DM) for 1 h at room temperature in sealed microcentrifuge tubes. Samples were dried under $N_2$ before LC-MS analysis.

UPLC conditions: Dried samples were reconstituted in 20-100 μl LC-MS grade methanol at room temperature. A Waters ACQUITY FTN autosampler held at 4 °C injected 2 μl onto the UPLC. For C4 separations, analytes were resolved on a Waters ACQUITY UPLC Protein BEH C4 column (1.7 μm, 1 × 100 mm, 300 Å) at 0.10 ml min$^{-1}$ (17-min run). Mobile phases were 10 mM formic acid in water (A) and 10 mM formic acid in acetonitrile (B). The gradient was: 67% B (1 min hold), ramp to 85% B over 9 min, hold 2 min, then re-equilibrate to the initial composition for 3 min. In initial C8 method trials, a Waters ACQUITY I-Class UPLC was operated at 0.30 ml min$^{-1}$ (15 min run) with 10 mM heptafluorobutyric anhydride (HFBA) in water (A) and 10 mM formic acid in acetonitrile (B), starting at 50:50 (v/v) for 1 min, followed by a linear ramp to 100% B over 9 min, a 4-min hold at 100% B, and a 1 min re-equilibration.

Mass spectrometry: Eluting compounds were analyzed on a Waters Xevo TQ-S triple quadrupole mass spectrometer operated in positive-ion electrospray and multiple reaction monitoring (MRM) mode for quantitative detection.

Quantification and normalization: Chromatographic peaks were integrated in MassLynx (Waters). For each phosphoinositide species, the area under the curve was normalized to the corresponding synthetic internal standard and further corrected for sample input by total protein content. Extraction efficiency was monitored via the 37:4 PIP standard added pre-extraction to enable run-to-run comparability.

## Neuronal isolation

Animal handling and experiments were conducted following procedures approved by the UC Davis Institutional Animal Care and Use Committee (protocol # 22644). Pregnant C57BL/6 mice were purchased from The Jackson Laboratory and primary cortical neurons were isolated from E15-E18 mouse embryos via Papain dissociation (Worthington, Cat #LK003150). Briefly, dissected cortical tissues were digested in papain solution for 20 min at 37 °C followed by mechanical dissociation through pipetting. Neurons were then pelleted and resuspended in Earls Balanced Salt Solution (EBSS) supplemented with ovomucoid and DNase I. After 30 s centrifugation at 420 g, cortical neurons were plated on poly-D-lysine-coated coverslips at a density of $6 × 10^5$ cells / well and cultured at 37 °C and 5% $CO_2$ in Neurobasal medium (Gibco, Cat #21103-049) containing 2% B27, 1% GlutaMAX and 0.2% penicillin/streptomycin. Experiments were typically performed after 5–10 days in vitro (DIV).

## Immunofluorescence labeling

Cell culture medium was replaced with Phosphate Buffer Saline (PBS) solution and cells were fixed for 10 min at room temperature with 4% PFA. After 3 ×5 min washes, cells were incubated for 1 h at room temperature in a blocking solution containing PBS with 50% SEAblock buffer and 0.5% Triton X-100, followed by overnight incubation at 4 °C in PBS with 20% SEAblock buffer, 0.5% Triton X-100 and the following primary antibodies: rabbit polyclonal anti-DHHC3 (1:50, Abcam), mouse monoclonal (IgG1) anti-PI4KIIα (1:50, Santa Cruz), rabbit polyclonal anti-Lamp1 (1:100, Thermo Fisher), rabbit polyclonal anti-TGN46 (1:300, NovusBio), mouse monoclonal (IgG1) anti-Lamp1 (1:10, Abcam), rabbit monoclonal anti-mTOR (1:200, Cell Signaling Technology), Rabbit monoclonal ORP1L (1:200, Abcam) and rabbit recombinant monoclonal anti-Drp1 (1:250, Abcam). Following primary antibody incubation, cells were washed 5 × 5 mins in PBS and incubated for 1 h at room temperature with the following secondary antibodies: Alexa Fluor 488 Goat anti-rabbit (1:1000, Invitrogen), Alexa Fluor 647 Goat anti-rabbit (1:1000, Invitrogen), Alexa Fluor 488 Goat anti-mouse IgG1 (1:1000, Invitrogen) and Alexa Fluor 647 Goat anti-mouse IgG1 (1:1000, Invitrogen). Finally,

cells were washed 5×5 mins in PBS and imaged at room temperature, using a Zeiss LSM880 AiryScan confocal microscope.

## Microscopy and image analysis

Live and fixed cells were imaged at room temperature in Ringer's solution (160 NaCl, 2.5 KCl, 2 CaCl$_2$, 1 MgCl$_2$, 10 HEPES, and 8 D-Glucose) and Phosphate-buffered saline (PBS), respectively.

Super-resolution confocal microscopy: Images were acquired using a Zeiss LSM880 laser scanning confocal microscope equipped with a Plan-Apochromat 63×/1.4 NA oil immersion objective. Samples were imaged using multiple optical sections at 0.5 μm depth intervals (Z-stacks). Channels were acquired sequentially using appropriate filters and excitation lines (488, 514, 594, and 633 nm for GFP/Alexa-488, YFP, mCherry/RFP, and Alexa-647, respectively). Images were processed using the Airyscan processing toolbox in ZEN software.

Spinning Disk confocal microscopy: Live cells were imaged using an Andor W1 spinning-disk confocal microscope coupled to a Photometrics Prime 95B camera. RFP/mCherry and GFP were visualized using 561 nm and 488 nm excitation wavelengths, respectively. Images were acquired through a Plan-Apochromat 60×/1.49 NA oil objective using Micromanager software.

Image Analysis: Unless otherwise specified, image analysis was performed using ImageJ (NIH, https://fiji.sc). Z-stack images underwent maximum intensity projection, followed by background subtraction using a 10–20 pixel rolling ball radius. Fluorescence signals were quantified within regions of interest focused on the perinuclear region or endomembranes, and normalized to cytosolic fluorescence intensity. For colocalization analysis, P4M, PI4KIIα, mTOR, and ATG9A signals were thresholded and measured within masks generated from Lamp1 or TGN46 channels.

## Bioluminescence Resonance Energy Transfer (BRET) imaging

HEK293t cells were seeded on coverslips 48 h prior to imaging and transfected with AIMTOR biosensor plasmid constructs[41] 24 h later. Thirty minutes before imaging, the cell culture medium was replaced with 500 μL of BRET recording medium: phenol red-free DMEM (Gibco, Cat #31053-028) supplemented with 4 mM L-Glutamine, 4.5 g/L D-glucose, 1 mM Sodium Pyruvate and 10% FBS. After adding 25 μl of Nanoluciferase substrate (Nano-Glo Live Cell Assay System, Progema), luminescence from YPET and Nanocuciferase was collected at 530 nm and 460 nm, respectively, using an Andor W1 spinning-disk confocal microscope equipped with a Plan-Apochromat 60×/1.49 oil objective and connected to a Photometrics Prime 95B camera. BRET signal analysis was performed using Image J with the BRET-Analyzer plugin[71]. Briefly, the background was subtracted on both channels and the 460 nm image underwent composite thresholding, followed by pixel-by-pixel division of 530 nm / 460 nm channels. The resulting ratio intensity image was displayed on a pseudocolor scale.

## Protein extraction and immunoblot

HAP1 cells were lysed in RIPA buffer (1% Triton X-100, 0.5% sodium deoxycholate, 0.1% SDS, 150 mM NaCl, 1 mM EDTA, 20 mM HEPES [pH 7.4]) supplemented with COmplete Mini Protease Inhibitor cocktail (Roche). Protein concentrations were determined using a Pierce BCA assay (Thermo Fisher Scientific). Lysates were denatured at 70 °C for 10 min, and proteins were resolved by SDS-PAGE onto 4–12% Tris-glycine mini-gels (Thermo Fisher Scientific). Proteins were then transferred to PVDF membranes using a Mini-Bolt system (Thermo Fisher Scientific). Transfer efficiency was verified by Ponceau S staining. Membranes were blocked 1 h at room temperature in Tris-buffered saline (TBS) containing non-fat dry milk 5% and Tween-20 0.1% then incubated overnight at 4 °C with anti-PI4KIIα mAb (1:100 dilution, Santa Cruz Biotechnology), anti-PI4KIIIβ mAb (1:1000 dilution, BD Biosciences), anti-Phospho-ULK1 (1:1000 dilution, Cell Signaling

Technology), anti-ULK1 (1:1000 dilution, Cell Signaling Technology), anti-Phospho-4E-BP1 (1:1000 dilution, Cell Signaling Technology), anti-4E-BP1 (1:1000 dilution, Cell Signaling Technology), and anti-GAPDH (1:10000 dilution, Proteintech) and anti-β-actin mAb (1:10000 dilution; Thermo Fisher Scientific) for normalization. After three washes (10 min each) with 0.1% Tween 20–containing TBS solution, membranes were incubated for 1 h at room temperature with Goat anti-Mouse 800CW fluorescent secondary antibody (1:10000 dilution, LI-COR Biosciences) or with horseradish peroxidase-conjugated Mouse anti-Rabbit secondary antibody (Jackson Immunoresearch). Immunoreactive bands were visualized on Sapphire Gel Imager (Azure Biosystems) or developed using Prometheus ProSignal Femto (Genesee Scientific, El Cajon, CA, USA) on autoradiography film (Amersham Hyperfilm ECL, Cytivia, Global Life Sciences Solutions USA, LLC., Marlborough, MA, USA) using a film developer; and subsequently quantified using the BioImporter plugin for ImageJ software.

### Cholesterol labeling
Cells were fixed for 10 min in PBS containing 3% paraformaldehyde and 0.1% glutaraldehyde, then treated with 0.1% sodium borate ($NaBH_4$) in $H_2O$ for 5 min. After washing, cells were incubated for 2 h in PBS with 0.05 mg/ml filipin. Following Filipin incubation, cells were washed three times for 5 min in PBS. Filipin was imaged using a Zeiss LSM880 AiryScan confocal microscope with 405 nm excitation.

### Lysosomal damage assay
HEK293t cells were transfected with the lysosomal damage marker Gal3-mCherry. DMSO or Apilimod treatment was applied to culture media 2 h prior to imaging. Cells were then perfused with a Ringer's solution containing (in mM) 160 NaCl, 2.5 KCl, 2 $CaCl_2$, 1 $MgCl_2$,10 HEPES, and 8 D-Glucose. After 5 min of baseline acquisition, the medium was replaced with Ringer's solution supplemented with 500 μM L-leucyl-L-leucine methyl ester (LLOME) for 10 min, followed by replacement with standard Ringer's solution for 2 h. Images were acquired at room temperature with a frequency of 1 frame per minute using an Andor W-1 spinning disk confocal microscope equipped with a Photometrics Prime 95B camera.

### Lysosomal acidification assay
HEK293t cells were transfected with the lysosomal-targeted, pH-sensitive construct LAMP1-pHluorin. Two hours prior to imaging, cells were exposed to DMSO or 1 μM Apilimod treatments, while 500 μM LLOME treatment was applied for 30 min only. Cells were then perfused with a Ringer's solution containing (in mM) 160 NaCl, 2.5 KCl, 2 $CaCl_2$, 1 $MgCl_2$,10 HEPES, and 8 D-Glucose. After a 100 s baseline acquisition, the medium was replaced with Ringer's solution supplemented with $NH_4Cl$ (pH 8) and maintained for the remainder of the recording. pHluorin fluorescence was imaged at room temperature using an Andor W-1 spinning disk confocal microscope equipped with a Photometrics Prime 95B camera, at a rate of 1 frame every 5 s.

### Transmission electron microscopy
HEK293t cells were seeded on 10 cm dishes and treated with DMSO (control) or 1 μM Apilimod for 2 h at 37 °C. Cells were harvested, resuspended in 1 mL of 0.1 M phosphate buffer (pH 7.4), and centrifuged for 5 min at 1000 g. The supernatant was removed and replaced with 1 mL of fixative solution (phosphate buffer supplemented with 2.5% glutaraldehyde and 2% paraformaldehyde). Following fixation, cells were washed three times with phosphate buffer (10 min each) and post-fixed with 1% osmium tetroxide containing 1% potassium ferricyanide for 1 h at 4 °C. Cells were dehydrated through a graded ethanol series (30%, 50%, 70%, 90% ethanol for 10 min each, followed by three changes of 100% ethanol for 15 min each) and infiltrated with Epon resin through progressive resin:ethanol mixtures. After embedding in fresh Epon and polymerization (overnight at 37 °C,

then 48 h at 60 °C), ultrathin sections (70-80 nm) were cut using an ultramicrotome and collected on copper grids. Sections were stained with uranyl acetate and lead citrate, then examined using a FEI Thermo Scientific Talos L120C Transmission Electron Microscope equipped with CETA 16 MP camera. Three-way membrane contact sites between ER, lysosomes, and mitochondria were identified as regions where all three organelles were within 30 nm of each other with visible membrane apposition and electron-dense material at contact interfaces.

### Mitochondrial morphology analysis
Live cells were incubated for 30 min at room temperature in 2 mM $CaCl_2$ Ringer's solution containing 100-200 nM MitoTracker Deep Red FM dye (Thermo Fisher Scientific). Cells were imaged using a Zeiss LSM880 AiryScan confocal microscope with 633 nm excitation. Images were processed using Zeiss Zen software. Mitochondrial length and branching were quantified using the "Filament Tracer" module in Imaris (BitPlane).

### Mitochondrial respiration assay
Mitochondrial oxygen consumption rate (OCR) was measured using a Seahorse XF96 analyzer (Agilent). HEK293t cells were seeded on Seahorse cell culture microplates at $1 \times 10^4$ cells/well and treated overnight with either DMSO (control) or 1 μM YM201636. Two hours before the assay, the cell culture medium was replaced with Seahorse XF DMEM (supplemented with 1 mM sodium pyruvate, 10 mM D-glucose, 2 mM L-glutamine, pH 7.4), and 1 μM SBI-0206965 was added. The XF96 sensor cartridge injection ports were loaded with oligomycin, FCCP, and rotenone/antimycin A to achieve final well concentrations of 1.5 μM, 0.25 μM, and 0.5 μM, respectively. OCR was measured under basal conditions and in response to the sequential addition of these compounds. Cells were subsequently lysed in RIPA buffer with protease inhibitors, and protein content was quantified by BCA assay for data normalization.

### Statistical analysis
Data analyses were performed using Microsoft Excel and GraphPad Prism softwares. All data are presented as mean ± s.e.m and single data points on graphs represent values for individual cells, except for the BRET assay and Seahorse data, where individual points represent average values per field of view and per well, respectively. Statistical significance was determined using one-way ANOVA and two-way ANOVA for multiple comparisons, after verifying the normality of distribution using a D'Agostino & Pearson test and identifying potential outliers using a two-sided Grubbs' test. $p$-values under 0.05 were considered statistically significant and are indicated in red.

### Reporting summary
Further information on research design is available in the Nature Portfolio Reporting Summary linked to this article.

## Data availability
The data supporting the findings of this work can be found in the source data file. The lipidomic data generated in this study have been deposited in the MetaboLights database under accession code: MTBLS13254. Source data are provided with this paper.

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

## Acknowledgements

We thank members of the Dickson and Dixon laboratories for helpful advice and constructive comments during this study. The authors are extremely grateful to those laboratories that shared reagents, plasmids, and cells lines used in this study. We thank Bradley Shibata from the Biological Electron Microscopy Facility at UC Davis. This work was supported by NIH grants R01 GM127513, R35 GM149211, and RF1 NS131379 (E.J.D.), and NIH grant R01 AG063796 (R.E.D). Illustrations were generated using Biorender.com.

## Author contributions

C.K., M.C., and E.J.D. performed experiments. C.K and E.J.D. performed image analysis. C.K., M.C., R.E.D., and E.J.D. designed experiments. All authors contributed to the writing of the manuscript.

## Competing interests

The authors declare no competing interests.
