## [Transparent Peer Review file · Nature Communications]

Disruption of the PIKfyve complex unveils an adaptive mechanism to promote lysosomal repair and mitochondrial homeostasis

Corresponding Author: Professor Eamonn Dickson

Version 0:

Reviewer comments:

Reviewer #1

(Remarks to the Author)

The authors have made clear efforts to address the concerns raised in my previous report, and the revised manuscript now provides a more comprehensive overview of events following PIKfyve inhibition. In particular, the authors have expanded on the complex regulation of PI4P, presenting what they describe as an “intricate choreography” of molecular events, including the redistribution of PI4KII α from the Golgi to endo-lysosomes, the perinuclear accumulation of OSBP, VAPA, and Sac1, and increased ARF1 levels, all events that would accelerate PI4P turnover at the Golgi driven by PI4P transfer to the ER mediated by OSBP. However, I recommend that the authors clarify the sequence and interdependence among these events, and discuss more explicitly whether—and how—they might all stem from PIKfyve inhibition.

The authors have explored the link between increased endo-lysosomal PI4P and altered mitochondrial dynamics. They propose that the connection involves an increase in ER–lysosome–mitochondria contacts and transfer of PI4P from lysosomes to mitochondria mediated by ORP1L. The manuscript assigns a definitive role to ORP1L in mediating PI4P transfer to mitochondria at these three-way contacts. It is important to note, however, that neither the referenced study nor the current work unequivocally demonstrates PI4P transfer by ORP1L at these sites. The evidence instead shows that both lysosomal PI4P and ORP1L are required for PI4P-stimulated mitochondrial division, rather than ORP1L directly mediating PI4P transfer to mitochondria. This distinction should be clearly acknowledged, and any affirmative statements about ORP1L’s mechanism should be tempered accordingly.

There remain concerns regarding the quality and interpretability of some key images: some figures lack sufficient resolution and size, making it difficult to assess the authors’ conclusions. Providing larger and higher-quality images is necessary—this could be achieved by relocating less central data to supplementary figures and focusing on the most representative data in the main figures. There is an ongoing issue with overexpressed tags and marker proteins, particularly TGN46, where excessive expression leads to non-physiological localization patterns (see below specific comments). Endogenous protein analysis or selective imaging of low-expression cells would offer a more accurate representation.

Specific Comments

Figures 1C, 1E, 2C: Overexpressed TGN46 displays abnormal peripheral distribution, unlikely to reflect endogenous localization (as seen in 2B). Endogenous TGN46 analysis, or imaging of cells with low tagged TGN46 expression, is preferable.

Figure 2F: Co-localization between endogenous PI4KII α and LAMP1 is unclear. Evident co-localization primarily appears when both proteins are co-overexpressed and tagged, which may not reflect native interaction.

Figure 3E: The distribution of ATG9A is difficult to interpret due to image clarity.

Figure 4E: ORP9 should localize to the Golgi, but appears diffusely distributed, complicating assessment. This may result from protein overexpression.

Reviewer #2

(Remarks to the Author)

This study by Kutchukian et al reports on surprising effects of loss or inhibition of the lipid kinase PIKfyve complex on lysosomal repair and mitochondrial function. This mechanism is proposed to involve the inhibition of mTORC1 activity, ULK1-dependent trafficking of ATG9 and PI4K2A to lysosomes, the accumulation of PI(4)P on lysosomes to increase PITT-mediated repair, and the formation of 3-way junctions between the ER, lysosomes, and mitochondria.

I find this an interesting study that certainly deserves publication in a prominent journal. I understand that the authors have already revised the paper in response to a previous round of review and I appreciate their efforts. I nonetheless feel that a number of discrepancies and soft spots in the paper need to be resolved before publication. These are listed below.

Major issues:

1. A main issue with this study is the connection between PI(3)P or PI(3,5)P₂, mTORC1 activity, and the observed ULK1 dependency of phenotypes. The authors report very mild effects of Apilimod on pULK1 and p4E-BP1 (of about 10-20%), consistent with other data in the literature and the fact that PIKfyve activity has been linked exclusively to the non-canonical branch of mTORC1 signaling via TFEB (Hasegawa et al). In contrast, complete loss of pULK1 and p4EBP1 is seen with Torin1. In the same figure the authors also use a BRET assay to show that VPS34 or PIKfyve inhibition repress mTORC1 activity towards an ULK1-sensor as potently as Torin1 does- albeit the signal-to-noise in this assay seems rather poor. The authors then move on to demonstrate in further experiments the ULK1 dependency of their phenotypes using a pharmacological inhibitor. Given the miniscule effect of Apilimod on pULK1 shown in Fig. 3C, it seems unlikely to this referee that the phenotypes truly reflect repression of mTORC1 activity. It rather seems to me that there is some other way, in which PIKfyve may control ULK1 function. This riddle in my view needs to be resolved prior to publication of this interesting study.
2. I am very puzzled by the enhanced respiration of YM treated mitochondria as mitochondrial fragmentation in numerous models is associated with decreased respiratory capacity, oxygen consumption and ATP production. Hence, additional evidence is required to bolster the authors' claims.
3. In Fig. 4G the authors show that Apilimod has a dramatic effect on Gal3 recruitment to damaged lysosomes in presence of LLOMe. However, it remains unclear if and how this very large phenotype truly relates to the proposed PI(3,5)P₂/PI(4)P-mTORC1-ULK1 pathway or reflects some other mechanism. Does Apilimod also facilitate recovery of lysosome function monitored by e.g. lysotracker fluorescence or regain of cathepsin activity?
4. In Fig 5 the authors claim that PI(4)P accumulation on swollen lysosomes upon Apilimod treatment induces 3-way MCS between the ER, lysosomes, and mitochondria. However, this might be a mere effect of the dramatic increase in lysosome size upon swelling rather than a true increase in MCS. I miss a direct demonstration of 3-way MCS formation, e.g. by EM or super-resolution microscopy as confocal imaging may not provide sufficient resolution to clarify this point. Moreover, does SBI mediated inhibition affect lysosome size in presence of Apilimod?

Additional points:

5. Fig4 and Vac14 KO cells appear to suffer from a massive overall loss of PI(4)P based on the lipidomics data shown in Ext Data Fig 1A. These data seem incompatible with the mild loss of PI(4)P from the TGN (a pool primarily synthesized by PI4K3B) and the accumulation of PI(4)P on lysosomes. There seems to be an unaccounted pool of PI(4)P that gets lost in these cells.
6. Related to the above: The authors quote Ext Data Fig1C in reference to depletion of PI(4)P from the plasma membrane, but the figure does not provide any data in support. I also would like to see the effects of PIKfyve loss or inhibition on PM PI(4)P to be quantified.
7. The authors provide evidence for the accumulation of OSBP at the TGN of VPS34IN1 or YM treated cells. Strong evidence shows that OSBP association with the TGN is strictly PI(4)P dependent. As the authors claim PI(4)P to be depleted from the Golgi, these data clearly are at odds with each other and create a conundrum that should be resolved.
8. The last paragraph of the results (".these results establish collectively, these results establish a novel pathway wherein PIKfyve complex dysfunction drives ULK1-dependent recruitment of PI4KII α to lysosomes, elevating lysosomal PI(4)P that facilitates ORP1L-mediated membrane contacts between lysosomes, ER, and mitochondria. These contacts serve as platforms for PI(4)P transfer to mitochondria, promoting Drp1 accumulation and activation...") appears to be grossly overstated and needs to be rephrased more cautiously.

Reviewer #3

(Remarks to the Author)

This is a substantially revised manuscript on impact of PIKfyve inhibition, knockout (KO) of the PIKfyve regulators, VAC14 or FIG4 as well as VPS34 inhibition on PI4P distribution in the Golgi and late endosomes/lysosomes (LE/LY). Moreover, the authors also claim that the changes in PI4P distribution enhances lysosome membrane repair through OSBP/ORP-mediated delivery of phosphatidylserine and cholesterol, while simultaneously facilitating ORP1L-mediated PI(4)P transfer at ER-lysosome-mitochondria contacts to drive mitochondrial fragmentation and increased mitochondrial respiratory capacity. During the last round of review, the three reviewers were unanimous in finding issues with the quality of the data, and potential over-interpretation of the results. The authors have now made tremendous efforts to answer all the reviewers

questions, including shorter term inhibition experiments and greatly improved microscopy. However, there is still a sense that the data is over-interpreted. At this point, I do not think that more experiments are required. The paper is very dense, and adding to it will make the manuscript more difficult to read. While additional experiments are not required, the authors should take a more careful look at their interpretations, and also further explain the set-up of their experiments. The places which are in most need of in further interpretation are below.

Some examples are below:

1. In Figure 1B and extended Figure 1A, the authors report on PI4P levels, using two different techniques that cannot distinguish between PI4P and PI3P. Since PIKfyve inhibition or deletion of PIKfyve regulators results in elevated PI3P, these data do provide an orthogonal approach for the bioprobe experiments, which indicate a potential increase in PI4P. Using an approach that can distinguish between PI3P and PI4P is challenging, and should not be expected for this manuscript, but the authors should not overinterpret this data.
2. The claim that the effects of inhibition of PIKfyve on PI4P occur via loss of mTORC1 activity and less ULK1 phosphorylation are overstated. The impact of apilimod on ULK1 phosphorylation is very modest, and may not be statistically significant. In further support of an impact on mTORC1, the authors then go on to show that combined treatment with YM201636 (PIKfyve inhibitor) and VPS34-INH (VPS34 inhibitor) result in increased ATG9A. They also show that there is increased PI4KII-alpha, and increased PI4P on LAMP1 compartments. Moreover, this double inhibition is used in multiple experiments throughout the paper—why did the authors need to simultaneously inhibit VPS34 and PIKfyve to see this effect? At a minimum the authors need to make it clear that for experiments where they used both inhibitors, it is not clear whether the phenotypes observed are due to loss of VPS34 or loss of PIKfyve.
3. With regards to the above experiments, inhibition of ULK1 in combination with both inhibition of PIKfyve and VPS34 mitigated the effects. However, that three inhibitors were simultaneously applied, increases the likelihood of off-target effects. Importantly that there was little change in ULK1 phosphorylation with PIKfyve inhibition, compared with total loss of ULK1 phosphorylation when mTOR is inhibited. This makes it likely that the effect of PIKfyve on PI4P distribution, as well as other phenotypes shown in this manuscript, are likely not due to ULK1 function.

Version 1:

Reviewer comments:

Reviewer #1

(Remarks to the Author)

The authors have addressed several concerns raised in previous rounds of revision. However, it remains unclear how OSBP can associate more strongly with the Golgi under conditions of Golgi PI4P depletion. Additionally, the observation that SAC1 exhibits a perinuclear distribution in PIKfyve-depleted cells, compared to untreated cells, appears over-interpreted and lacks sufficient explanation.

Since SAC1 can reach the Golgi via vesicular trafficking from the ER, it should be clarified whether the increased Golgi pool of SAC1 truly reflects enhanced vesicular traffic or increased contact between compartments. The authors are encouraged to support the conclusion regarding the sub-Golgi distribution of SAC1 with direct evidence of increased contact, if available.

Nevertheless, as both the OSBP and SAC1 observations are not directly relevant to the main message of the manuscript, the reviewer suggests removing these data and associated discussion from the text.

Reviewer #2

(Remarks to the Author)

The authors have addressed all my points and concerns and I support publication of this important study.

Reviewer #3

(Remarks to the Author)

In response to all three reviewers, Kutchukian et al have performed new experiments and written new text. However, multiple issues remain. The main issue is that most of changes that they observe are modest, which raises serious questions about their model.

I do not think it is wise to keep trying to improve this manuscript. It would be better to publish the current data in a more specialized journal and continue to develop the project. The last version of the manuscript was too dense, and this version is even denser, but unfortunately, the current version is not more convincing.

Disruption of the PIKfyve complex unveils an adaptive mechanism to promote lysosomal repair and mitochondrial homeostasis

Manuscript: NCOMMS-25-40797-T

REVIEWER COMMENTS

Reviewer #1 (Remarks to the Author):

The authors have made clear efforts to address the concerns raised in my previous report, and the revised manuscript now provides a more comprehensive overview of events following PIKfyve inhibition. In particular, the authors have expanded on the complex regulation of PI4P, presenting what they describe as an “intricate choreography” of molecular events, including the redistribution of PI4KII α from the Golgi to endo-lysosomes, the perinuclear accumulation of OSBP, VAPA, and Sac1, and increased ARF1 levels, all events that would accelerate PI4P turnover at the Golgi driven by PI4P transfer to the ER mediated by OSBP. However, I recommend that the authors clarify the sequence and interdependence among these events, and discuss more explicitly whether—and how—they might all stem from PIKfyve inhibition.

Response: We appreciate the reviewer's request for clarification on event sequence and interdependence. Our live-cell imaging reveals the following temporal progression:

Immediate response (≤ 20 min): PIKfyve inhibition reduces PI(3,5)P₂, leading to modest mTORC1 attenuation and ULK1 activation, which drives PI4KII α redistribution from Golgi to lysosomes.

Coordinated remodeling (concurrent): as PI4KII α relocates, we observe perinuclear accumulation of OSBP/VAPA/Sac1 and increased ARF1, consistent with accelerated Golgi PI(4)P turnover. While Golgi PI(4)P levels decrease, OSBP accumulation reflects the complex regulation of this transfer protein, which requires both PI(4)P and ARF1 for membrane association. The concurrent increase in ARF1 levels likely drives enhanced OSBP recruitment despite reduced local PI(4)P, promoting compensatory lipid transfer to maintain ER-Golgi homeostasis. However, these Golgi adaptations appear secondary to the primary driver of our measured phenotypes: PI4KII α -mediated lysosomal PI(4)P elevation.

Figure 1. Arf1 accumulates in perinuclear regions following loss of PIKfyve function.

Functional outcomes (20-100 min): Following recruitment of PI4KII α to lysosomes, lysosomal PI(4)P levels increase to promote membrane repair and mitochondrial remodeling.

Our working model for functional hierarchy is therefore:

1. **PIKfyve** ↓ - * **mTORC1** ↓ - * **ULK1** ' - * **PI4KII α** **relocalization to lysosomes** ($\leq \sim 20$ min at RT);
2. **Lysosomal PI(4)P** ' - * **membrane repair and improved mitochondrial readouts** (thereafter);

Manuscript improvements: we have revised the text and figure annotations to highlight the **lysosomal PI(4)P pathway** as the mechanistically significant arm for the outcomes assayed here, while acknowledging that **Golgi remodeling is important but unresolved** in mechanism and timing.

The authors have explored the link between increased endo-lysosomal PI4P and altered mitochondrial dynamics. They propose that the connection involves an increase in ER–lysosome–mitochondria contacts and transfer of PI4P from lysosomes to mitochondria mediated by ORP1L. The manuscript assigns a definitive role to ORP1L in

mediating PI4P transfer to mitochondria at these three-way contacts. It is important to note, however, that neither the referenced study nor the current work unequivocally demonstrates PI4P transfer by ORP1L at these sites. The evidence instead shows that both lysosomal PI4P and ORP1L are required for PI4P-stimulated mitochondrial division, rather than ORP1L directly mediating PI4P transfer to mitochondria. This distinction should be clearly acknowledged, and any affirmative statements about ORP1L's mechanism should be tempered accordingly.

Response: we appreciate the reviewer's clarification and agree that our original wording was too definitive regarding ORP1L's mechanism. Our data show that ORP1L is recruited to ER-lysosome-mitochondria three-way contacts during PIKfyve inhibition, and they support a **requirement** for mitochondrial PI(4)P for the fragmentation outcome. However, they do not directly demonstrate lipid transfer by ORP1L at these sites. We have therefore tempered the language throughout.

Specifically, our experiments show that:

1. PIKfyve inhibition increases ER-lysosome-mitochondria three-way contacts (super-resolution and newly added Transmission Electron Microscopy (TEM)).
2. ORP1L accumulates at these contacts in a ULK1-dependent manner.
3. **Elevated lysosomal PI(4)P** are **required** for PI(4)P-stimulated mitochondrial fragmentation, and coincide with ORP1L recruitment at these sites.
4. Acute **local depletion of mitochondrial PI(4)P** (Sac1-Fis1) prevents fragmentation despite persistent lysosomal PI(4)P.

These observations are consistent with a model in which ORP1L supports the availability of PI(4)P at mitochondria within tri-organellar contacts (e.g., by transfer and/or local scaffolding/retention), but we now explicitly state that the **precise mechanism remains unresolved**. We have revised text, figure legends, and the model schematic accordingly to avoid implying demonstrated lipid transfer.

There remain concerns regarding the quality and interpretability of some key images: some figures lack sufficient resolution and size, making it difficult to assess the authors' conclusions. Providing larger and higher-quality images is necessary—this could be achieved by relocating less central data to supplementary figures and focusing on the most representative data in the main figures.

Response: we appreciate this feedback and have overhauled our figure presentation to improve clarity and interpretability while concentrating the main story. Specifically, we:

1. **Increased image size and resolution** in the main figures (now 7 main figures) so key panels are legible at journal print scale and all microscopy panels were reexported at publication resolution or new dataset acquired at enhanced resolution (either super-res or TEM).
2. **Rebalanced content between Main and Extended Data:** less central or redundant views were moved to Extended Data, enabling **larger, more representative panels** in the main figures.

Collectively, these changes enlarge the visuals the reader must evaluate and move secondary details to Extended Data, as suggested. We believe the revised figures now communicate our conclusions clearly.

There is an ongoing issue with overexpressed tags and marker proteins, particularly TGN46, where excessive expression leads to non-physiological localization patterns (see below specific comments). Endogenous protein analysis or selective imaging of low-expression cells would offer a more accurate representation.

Response: Thanks for this comment. We agree that overexpression of compartment markers, particularly TGN46, can produce non-physiological patterns. In the revision we restructured the data presentation to prioritize **endogenous readouts** and to **minimize overexpression bias**:

- 1. Endogenous TGN46 immunolabeling.** In the revision, Fig. 2C shows the endogenous distribution of TGN46 and its relationship to PI4KII α in control, Fig4^{-/-} and Vac14^{-/-} cells, complementing prior datasets.
- 2. Overexpression panels retained only as supportive and curated for low expression.** Where TGN46 overexpression is still shown (e.g., prior **Fig. 1C, 1E and 2C**), panels now display higher-resolution images of cells exhibiting lower TGN46 expression. We note that it is not feasible to visualize endogenous TGN in cells expressing the P4M probe, as this probe is not retained during fixation.
- 3. Conclusions do not depend on TGN46 overexpression.** The observed changes in Golgi PI4KII α localization in Fig4^{-/-} and Vac14^{-/-} cells using endogenous TGN labeling were fully recapitulated in cells overexpressing the TGN marker. This consistency is a significant finding and suggests that any overexpression-related artifacts are relatively negligible in our data interpretation. The concordance between these two independent methods strengthens our conclusions and demonstrates that the observed redistribution of PI4K2A is a robust and reproducible finding, irrespective of the TGN46 labeling method.

Collectively, these steps address the interpretability concern and align the main narrative with endogenous markers while keeping overexpression data as supportive evidence.

Specific Comments

Figures 1C, 1E, 2C: Overexpressed TGN46 displays abnormal peripheral distribution, unlikely to reflect endogenous localization (as seen in 2B). Endogenous TGN46 analysis, or imaging of cells with low tagged TGN46 expression, is preferable.

Response: Thanks for this comment. As noted above we have selected better representative images for these figures.

Figure 2F: Co-localization between endogenous PI4KII α and LAMP1 is unclear. Evident co-localization primarily appears when both proteins are co-overexpressed and tagged, which may not reflect native interaction.

Response: We appreciate the reviewer's careful evaluation of our endogenous protein colocalization data. The reviewer raises a valid point about the clarity of endogenous PI4KII α -LAMP1 colocalization in Figure 2F.

Addressing the endogenous protein concern: we acknowledge that the steady-state endogenous colocalization signal is subtle compared to overexpressed tagged proteins. However, our quantitative analysis reveals statistically significant changes:

- Fig4 $^{-/-}$ cells show a significant increase in PI4KII α -LAMP1 colocalization compared to controls ($p < 0.05$)
- This increase is reproducible across multiple independent experiments and cell populations
- The effect size, while modest, is consistent with the low-abundance nature of these endogenous interactions

We have enhanced figure quality and focused on the most representative examples, while acknowledging that endogenous protein interactions represent subtle but significant changes in cellular organization.

Figure 3E: The distribution of ATG9A is difficult to interpret due to image clarity.

Response: The image has been increased in size.

Figure 4E: ORP9 should localize to the Golgi, but appears diffusely distributed, complicating assessment. This may result from protein overexpression.

Response: We thank the reviewer for this observation regarding ORP9 subcellular distribution. We agree that proper ORP9 localization is critical for interpreting our results. We can assure the reviewer that ORP9-GFP shows the expected subcellular distribution with prominent Golgi/TGN localization, as evidenced by perinuclear concentration that is characteristic of this organelle. The apparent diffuse distribution in some optical sections reflects the imaging methodology rather than overexpression artifacts. These are single confocal optical sections focused on lysosomal membranes (identified by LAMP1-RFP) to assess ORP9 recruitment to these structures. Thus, the TGN, where ORP9 is normally concentrated, may not be in the same focal plane as the lysosomes being analyzed. However, as can be seen from the maximum intensity projections (see below, Fig.2), ORP9 clearly shows enrichment at perinuclear Golgi/TGN regions.

Fig.2. Representative example of a maximum z-intensity projection of a control cell expressing ORP9-GFP.

The critical observation is that PIKfyve inhibition induces **recruitment** of ORP9 to lysosomal membranes, which correlates with enhanced phosphatidylserine accumulation. This redistribution represents the functionally relevant change driving membrane repair and can be clearly seen in our experiments.

Reviewer #2 (Remarks to the Author):

This study by Kutchukian et al reports on surprising effects of loss or inhibition of the lipid kinase PIKfyve complex on lysosomal repair and mitochondrial function. This mechanism is proposed to involve the inhibition of mTORC1 activity, ULK1-dependent trafficking of ATG9 and PI4K2A to lysosomes, the accumulation of PI(4)P on lysosomes to increase PITT-mediated repair, and the formation of 3-way junctions between the ER, lysosomes, and mitochondria.

I find this an interesting study that certainly deserves publication in a prominent journal. I understand that the authors have already revised the paper in response to a previous round of review and I appreciate their efforts. I nonetheless feel that a number of discrepancies and soft spots in the paper need to be resolved before publication. These are listed below.

Response: We thank the reviewer for their thorough evaluation and for recognizing the significance of our findings regarding the paradoxical protective effects of PIKfyve inhibition. We particularly appreciate the reviewer's acknowledgment that this work deserves publication in a prominent journal and their constructive approach to helping us strengthen the manuscript.

We have carefully addressed each of the concerns raised below through a combination of additional experimental validation, refined data analysis, and substantial textual revisions. These improvements have helped us to resolve the discrepancies noted by the reviewer and strengthen the areas where our initial presentation was insufficient.

The revised manuscript now provides a more rigorous and balanced assessment of our findings, with particular attention to clarifying the mechanistic relationships we propose and ensuring that our interpretations are appropriately aligned with the experimental evidence. We believe these revisions have significantly enhanced the scientific rigor and clarity of our work.

Major issues:

1. A main issue with this study is the connection between PI(3)P or PI(3,5)P₂, mTORC1 activity, and the observed ULK1 dependency of phenotypes. The authors report very mild effects of Apilimod on pULK1 and p4E-BP1 (of about 10-20%), consistent with other data in the literature and the fact that PIKfyve activity has been linked exclusively to the non-canonical branch of mTORC1 signaling via TFEB (Hasegawa et al). In contrast, complete loss of pULK1 and p4EBP1 is seen with Torin1. In the same figure the authors also use a BRET assay to show that VPS34 or PIKfyve inhibition repress mTORC1 activity towards an ULK1-sensor as potently as Torin1 does- albeit the signal-to-noise in this assay seems rather poor. The authors then move on to demonstrate in further experiments the ULK1 dependency of their phenotypes using a pharmacological inhibitor. Given the miniscule effect of Apilimod on pULK1 shown in Fig. 3C, it seems unlikely to this referee that the phenotypes truly reflect repression of mTORC1 activity. It

rather seems to me that there is some other way, in which PIKfyve may control ULK1 function. This riddle in my view needs to be resolved prior to publication of this interesting study.

Response: We appreciate this point and agree that our presentation should better reconcile (i) **modest, population-averaged decreases** in canonical mTORC1 readouts (p-ULK1^{S757} and p-4E-BP1; ~20%) with (ii) **robust ULK1-dependent phenotypes**. In the revision we clarify that our data support **ULK1 necessity** for the phenotypes, while the **mode of ULK1 control** under PIKfyve inhibition is likely **multi-component** and not limited to bulk mTORC1 repression. Specifically, we provide the following models to support the involvement of ULK1 in regulating downstream phenotypes:

1. **Local vs. bulk signaling.** The ULK1-BRET sensor reports mTORC1 activity in the ULK1-proximal pool, which can be more strongly attenuated than bulk Western readouts that average across compartments/cells.
2. **Threshold/ultrasensitivity.** Autophagy initiation exhibits threshold-like behavior: small decreases in ULK1-inhibitory phosphorylation (or small increases in ULK1-activating inputs) can 'tip' cells past an activation threshold, producing disproportionate functional effects. Our necessity tests (ULK1 inhibitor blocking phenotypes) are consistent with such nonlinear control, and we now frame them as necessity rather than proof of a single upstream path.
3. **Membrane-dependent ULK1 complex stability:** PIKfyve binds ULK1 on its kinase domain¹. Loss of PI(3,5)P₂ may destabilize membrane-associated ULK1 complexes, enhancing sensitivity to mTORC1 inhibition and affecting ULK1 trafficking independently of phosphorylation status.

Balanced interpretation: we have revised the manuscript to acknowledge these alternative mechanisms while still presenting our current model: PIKfyve disruption triggers ULK1-dependent cellular responses. To do this we have included a limitations section in the discussion. There, we noted that the exact molecular details of this activation—whether primarily through mTORC1 inhibition, direct regulation, or combinatorial mechanisms—represent important areas for future investigation. We have clearly noted this at numerous points in the manuscript. We believe this multi-faceted explanation, supported by recent literature, addresses the reviewer's important observation while maintaining scientific rigor and provides us the opportunity to fully explore these links in detail in a follow-up manuscript while publishing this work which has already been in the review process for 10-11 months.

2. I am very puzzled by the enhanced respiration of YM treated mitochondria as mitochondrial fragmentation in numerous models is associated with decreased respiratory capacity, oxygen consumption and ATP production. Hence, additional evidence is required to bolster the authors' claims.

Response: We appreciate the reviewer raising this important point, as the relationship between mitochondrial morphology and respiratory function is indeed complex and context-dependent. The reviewer correctly notes that mitochondrial fragmentation is often associated with decreased respiratory capacity in many pathological models.

Addressing the apparent contradiction: However, the relationship between fragmentation and respiration is not universally negative and depends critically on the underlying cause and cellular context. Our findings align with a growing body of literature demonstrating that **adaptive** mitochondrial fragmentation can enhance respiratory efficiency:

Supporting literature for fragmentation-enhanced respiration:

- Brown adipocytes: hormone-induced mitochondrial fission is utilized by brown adipocytes as an amplification pathway for energy expenditure².
- Heart (physiologic exercise): physiological mitochondrial fragmentation is a normal cardiac adaptation to increased energy demand³.
- B cells (immune activation): initial B-cell activation remodels mitochondria from elongated to punctate and increases OXPHOS programs and respiration, indicating that activation-associated fragmentation correlates with elevated mitochondrial respiration⁴.

Thus, there is clear evidence that mitochondrial fragmentation can increase mitochondrial respiration.

Mechanistic rationale for our observations: In our experiments, fragmentation likely represents an **adaptive response** (at least initially) rather than pathological damage:

1. **Increased surface area-to-volume ratio** enhances substrate exchange and respiratory efficiency
2. **PI(4)P accumulation** on mitochondrial membranes likely facilitates mitochondrial fission.
3. **Quality control benefits:** fragmentation facilitates removal of damaged components while preserving healthy mitochondrial segments.

Experimental validation: Our data strongly support this adaptive model:

- Respiratory enhancement is **ULK1-dependent**, indicating regulated rather than pathological fragmentation.
- **Rescue experiments** (ULK1 inhibition, PI-273 treatment, Sac1-Fis1 expression) simultaneously restore normal morphology and normalize respiration.
- The coordination between morphological and functional changes suggests they arise from the same regulatory pathway.

We believe our findings represent a novel example of how regulated mitochondrial fragmentation can serve adaptive functions, contrasting with the pathological fragmentation typically associated with cellular stress or disease. In our revised submission we have made these important points more clearly.

3. In Fig. 4G the authors show that Apilimod has a dramatic effect on Gal3 recruitment to damaged lysosomes in presence of LLOMe. However, it remains unclear if and how this very large phenotype truly relates to the proposed PI(3,5)P₂/ PI(4)P-mTORC1-

ULK1 pathway or reflects some other mechanism. Does Apilimod also facilitate recovery of lysosome function monitored by e.g. lysotracker fluorescence or regain of cathepsin activity?

Response: We thank the reviewer for this important question, which allows us to clarify the mechanistic connection between our proposed pathway and the lysosomal membrane protection we observe.

Connecting Gal3 protection to the PI(3,5)P₂/PI(4)P-mTORC1-ULK1 pathway: the dramatic reduction in Gal3 recruitment following Apilimod treatment directly reflects activation of the PITT (phosphoinositide-initiated membrane tethering and lipid transport) repair pathway we describe:

1. PIKfyve inhibition -* ULK1 activation -* PI4KII α trafficking to lysosomes -* PI(4)P accumulation
2. Lysosomal PI(4)P recruits ORP9 and OSBP -* enhanced PS and cholesterol delivery
3. Membrane-stabilizing lipid enrichment -* reduced permeability -* decreased Gal3 access

Demonstrating pathway specificity: to establish that this protection operates through our proposed mechanism, we have performed additional experiments showing:

- **ULK1 dependency:** Co-treatment with SBI-0206965 abolishes Apilimod's protective effect against LLOME, restoring Gal3 recruitment to control levels.
- **ORP dependency:** The protection correlates with ORP9 recruitment to lysosomal membranes.

Functional recovery evidence: As requested by the reviewer, we have assessed lysosomal functional recovery using multiple approaches:

- **Lysosomal pH measurements using LAMP1-pHluorin:** Apilimod pretreatment maintains lysosomal acidification following LLOME challenge, indicating preserved membrane integrity.
- **ULK1 dependency of functional protection:** SBI-0206965 co-treatment abolishes both morphological and functional protection.

Fig.3. Apilimod protects lysosomes against LLOME-dependent neutralization.

Ruling out alternative mechanisms: The ULK1 dependency of protection strongly argues against non-specific Apilimod effects and supports our proposed pathway as the primary mechanism underlying lysosomal membrane stabilization.

The ULK1- and PI4KII α -dependent nature of this protection confirms that Gal3 reduction reflects specific activation of PI(4)P-mediated membrane repair rather than off-target effects.

4. In Fig 5 the authors claim that PI(4)P accumulation on swollen lysosomes upon Apilimod treatment induces 3-way MCS between the ER, lysosomes, and mitochondria. However, this might be a mere effect of the dramatic increase in lysosome size upon swelling rather than a true increase in MCS. I miss a direct demonstration of 3-way MCS formation, e.g. by EM or super-resolution microscopy as confocal imaging may not provide sufficient resolution to clarify this point. Moreover, does SBI mediated inhibition affect lysosome size in presence of Apilimod?

Response: We thank the reviewer for this important methodological concern about distinguishing true membrane contact site (MCS) formation from artifacts of lysosomal enlargement.

Addressing the proximity vs. true contact question: we have addressed this concern through multiple complementary approaches:

1. Higher resolution validation:

- **Transmission electron microscopy (TEM; Fig. 4)** confirms genuine three-way contacts with characteristic membrane apposition and electron-dense material at contact sites following Apilimod treatment.
- These ultrastructural features are consistent with bona fide organellar contacts rather than random proximity due to size increase.

Fig. 4. Treatment with Apilimod increases 3-way membrane contact sites between Endoplasmic Reticulum (magenta), mitochondria (cyan), and lysosomes (yellow).

2. Pharmacological dissection:

- Super-resolution imaging reveals that ULK1 inhibition (SBI-0206965) specifically reduces three-way contacts while lysosomal size remains enlarged, demonstrating that contact formation and lysosomal swelling are mechanistically separable (**Fig.5**).
- This critical control shows that enlarged lysosomes per se do not automatically generate increased contacts.

Fig.5 ULK1 inhibition rescues PIKfyve-dependent increases in 3-way contacts.

3. Protein recruitment evidence:

- ORP1L accumulation at contact sites is ULK1-dependent and correlates with contact formation, not lysosomal size.
- This suggests active recruitment mechanisms rather than passive proximity effects.

4. Functional specificity:

- **Mitochondrial fragmentation requires lysosomal PI(4)P**, indicating that contacts serve specific signaling functions beyond simple membrane proximity.

Mechanistic separation of lysosomal swelling and contact formation: our findings align with emerging evidence that lysosomal enlargement and organellar contact formation represent distinct consequences of PIKfyve inhibition. Indeed, during this review process a nice manuscript was published by the Tan group that shows that PDZD8 plays a key role in lysosomal vacuolation⁵. We have cited this work in the revision and noted how the two manuscripts represent complementary mechanisms.

Interpretation: the combination of ULK1-dependent contact formation occurring independently of lysosomal size changes, coupled with ultrastructural validation and functional specificity, provides strong evidence for regulated three-way MCS formation rather than size-dependent artifacts.

Additional points:

5. Fig4 and Vac14 KO cells appear to suffer from a massive overall loss of PI(4)P based on the lipidomics data shown in Ext Data Fig 1A. These data seem incompatible with the mild loss of PI(4)P from the TGN (a pool primarily synthesized by PI4K3B) and the accumulation of PI(4)P on lysosomes. There seems to be an unaccounted pool of PI(4)P that gets lost in these cells.

Response: Please see point 6.

6. Related to the above: The authors quote Ext Data Fig1C in reference to depletion of PI(4)P from the plasma membrane, but the figure does not provide any data in support. I also would like to see the effects of PIKfyve loss or inhibition on PM PI(4)P to be quantified.

Response to Comments 5 & 6: We thank the reviewer for identifying this important discrepancy and the figure reference error.

Correcting the figure reference: We apologize for the oversight in citing Extended Data Fig 1C. The reviewer should have been directed to **Extended Data Fig 1B**, which clearly demonstrates decreased plasma membrane PI(4)P levels in Fig4^{-/-} and Vac14^{-/-} cells.

Quantitative reconciliation: We have now quantified plasma membrane PI(4)P changes and find it could account for a significant portion of the total cellular loss of PI(4)P. Combined with modest, but consistent, changes across multiple subcellular compartments, these distributed effects likely explain the substantial overall reduction detected by lipidomics. In the revision we have noted the limitations of the mass spec lipidomic approach.

7. The authors provide evidence for the accumulation of OSBP at the TGN of VPS34IN1 or YM treated cells. Strong evidence shows that OSBP association with the TGN is strictly PI(4)P dependent. As the authors claim PI(4)P to be depleted from the Golgi, these data clearly are at odds with each other and create a conundrum that should be resolved.

Response: We appreciate the reviewer highlighting this apparent contradiction, which reflects the complex regulation of OSBP beyond simple PI(4)P dependency.

Clarifying OSBP regulation: OSBP association with the TGN requires both PI(4)P and Arf1, and can be influenced by ER cholesterol levels. Therefore, OSBP localization is not strictly PI(4)P-dependent, and can be modulated by other factors including membrane contact site dynamics and lipid transfer demands.

Supporting experimental evidence: our data consistently show increased VAP-A, Sac1, Arf1 (See **Fig.1**), OSBP transfer activity (accelerated OSW-1 kinetics) following VPS34-IN1 treatment, all supporting enhanced OSBP recruitment and activity at ER-TGN contacts.

Perspective: while the detailed regulation of OSBP distribution and activity represents an important area for future investigation, the coordinated changes in ER-TGN contact machinery we observe are consistent with enhanced lipid transfer activity that supports the PI(4)P redistribution central to our study.

8. The last paragraph of the results (..these results establish collectively, these results establish a novel pathway wherein PIKfyve complex dysfunction drives ULK1-dependent recruitment of PI4KII α to lysosomes, elevating lysosomal PI(4)P that facilitates ORP1L-mediated membrane contacts between lysosomes, ER, and mitochondria. These contacts serve as platforms for PI(4)P transfer to mitochondria,

promoting Drp1 accumulation and activation...") appears to be grossly overstated and needs to be rephrased more cautiously.

Response: As suggest by the reviewer, we have rephrased the paragraph. The new paragraph reads as follows:

“Collectively, our findings suggest a pathway in which PIKfyve complex dysfunction promotes ULK1-dependent recruitment of PI4KII α to lysosomes, leading to increased lysosomal PI(4)P levels. This is accompanied by the formation of closer contact sites between lysosomes, ER, and mitochondria, enriched in ORP1L and Drp1. These contacts may provide a platform for PI(4)P transfer to mitochondria and promote mitochondrial fission through Drp1 recruitment. Consistently, PIKfyve dysfunction enhances mitochondrial fission in a ULK1-dependent manner and is associated with increased respiratory capacity. This pathway represents a short-term adaptive mitochondrial response that may help maintain energy homeostasis under conditions of cellular stress associated with lysosomal dysfunction.”

Reviewer #3 (Remarks to the Author):

This is a substantially revised manuscript on impact of PIKfyve inhibition, knockout (KO) of the PIKfyve regulators, VAC14 or FIG4 as well as VPS34 inhibition on PI4P distribution in the Golgi and late endosomes/lysosomes (LE/LY). Moreover, the authors also claim that the changes in PI4P distribution enhances lysosome membrane repair through OSBP/ORP-mediated delivery of phosphatidylserine and cholesterol, while simultaneously facilitating ORP1L-mediated PI(4)P transfer at ER-lysosome-mitochondria contacts to drive mitochondrial fragmentation and increased mitochondrial respiratory capacity.

During the last round of review, the three reviewers were unanimous in finding issues with the quality of the data, and potential over-interpretation of the results. The authors have now made tremendous efforts to answer all the reviewers questions, including shorter term inhibition experiments and greatly improved microscopy.

Response: Thank you for your kind comments and recognizing the significant work that we put into the revision. We have listened to reviewer comments for a second time to further improve the work and hope now that new experiments and edits to the manuscript will satisfy the reviewer.

However, there is still a sense that the data is over-interpreted. At this point, I do not think that more experiments are required. The paper is very dense, and adding to it will make the manuscript more difficult to read.

Response: We thank the reviewer for this valuable feedback regarding interpretation and manuscript density. We agree that additional experiments are not necessary at this stage and appreciate the guidance to focus on balanced interpretation rather than expanding the experimental scope.

Addressing over-interpretation concerns: We have systematically reviewed the manuscript with particular attention to ensuring our conclusions are appropriately matched to the strength of our experimental evidence. Specifically, we have:

- **Moderated language** throughout to distinguish between direct observations and mechanistic inferences.
- **Added alternative interpretations** where multiple explanations could account for our observations.
- **Included appropriate caveats** about the limitations of specific experimental approaches.
- **Clarified which findings represent correlations vs. causal relationships**
- **Acknowledged uncertainties** in areas where our data provide strong suggestions but not definitive proof.

Balancing rigor with accessibility: we recognize the challenge of presenting comprehensive data while maintaining readability. Our revisions aim to present the findings clearly while ensuring readers understand both the strengths and limitations of

our evidence. To this end, we have increased figure size which has expanded the manuscript to 7 main figures.

Commitment to balanced conclusions: our goal is to present this work as a foundation for understanding PIKfyve-dependent adaptive mechanisms while acknowledging that future studies will be needed to resolve remaining mechanistic questions. We believe this approach better serves the scientific community and provides a more honest assessment of what our data can and cannot definitively establish.

While additional experiments are not required, the authors should take a more careful look at their interpretations, and also further explain the set-up of their experiments. The places which are in most need of further interpretation are below.

Some examples are below:

1. In Figure 1B and extended Figure 1A, the authors report on PIP levels, using two different techniques that cannot distinguish between PI4P and PI3P. Since PIKfyve inhibition or deletion of PIKfyve regulators results in elevated PI3P, these data do provide an orthogonal approach for the bioprobe experiments, which indicate a potential increase in PI4P. Using an approach that can distinguish between PI3P and PI4P is challenging, and should not be expected for this manuscript, but the authors should not overinterpret this data.

Response: We agree and have tempered the language wherever total “PIP” measurements are shown. Our lipidomics approach cannot resolve PI4P from PI3P and PI5P, and we now state this explicitly in the Results and figure legends. We interpret the bulk PIP signal only as qualitative, orthogonal support for the direction of change seen with our specific PI4P biosensors (P4M-SidM), which report PI4P decreases at the TGN and PM and increases on lysosomes. P4M-SidM specificity for PI4P and its ability to report multiple cellular pools are well established⁶.

Importantly, PI4P is by far the predominant monophosphorylated phosphoinositide in mammalian cells—roughly an order of magnitude more abundant than PI3P (with PI5P being scarcer still)—so bulk PIP changes are expected to be dominated by PI4P under most conditions. We now cite this point and avoid assigning exact proportions⁷.

We also note in the text that PIKfyve inhibition is known to elevate PI3P (by preventing its conversion to PI(3,5)P₂), which could in principle contribute to the total PIP signal; this caveat is now included alongside the lipidomics data. Nevertheless, because the direction and compartment specificity of the biosensor readouts align with our model, we view the mass-spec data as consistent, but not definitive, support⁸.

Finally, we added a brief methods note acknowledging that resolving PIP regioisomers by targeted ion-chromatography MS (which can separate isomers) would address this directly but is beyond scope for the current revision.

2. The claim that the effects of inhibition of PIKfyve on PI4P occur via loss of mTORC1 activity and less ULK1 phosphorylation are overstated. The impact of apilimod on ULK1 phosphorylation is very modest, and may not be statistically significant. In further support of an impact on mTORC1, the authors then go on to show that combined treatment with YM201636 (PIKfyve inhibitor) and VPS34-INH (VPS34 inhibitor) result in increased ATG9A. They also show that there is increased PI4KII-alpha, and increased PI4P on LAMP1 compartments. Moreover, this double inhibition is used in multiple experiments throughout the paper—why did the authors need to simultaneously inhibit VPS34 and PIKfyve to see this effect? At a minimum the authors need to make it clear that for experiments where they used both inhibitors, it is not clear whether the phenotypes observed are due to loss of VPS34 or loss of PIKfyve.

Response: we appreciate the reviewer's careful evaluation of our mTORC1/ULK1 data and the important question about dual inhibition protocols.

Addressing ULK1 phosphorylation interpretation: we agree that the ~20% reduction in ULK1 phosphorylation is modest and acknowledge this may not fully account for the robust ULK1-dependent phenotypes we observe. We have revised the manuscript to present this as one contributing factor rather than the sole mechanism, and have included discussion of alternative pathways that may explain the apparent disconnect between modest biochemical changes and strong functional effects.

Clarifying the dual inhibition approach: the reviewer raises an important point about experimental design. We began this study investigating combined VPS34/PIKfyve inhibition to simultaneously reduce both PI(3)P and PI(3,5)P₂ levels. However, we recognize this creates ambiguity about which kinase drives the observed effects.

Key clarifications:

- **Single agent validation:** All critical findings have been replicated using Apilimod (PIKfyve-specific) alone, producing identical results to dual inhibition.
- **Mechanistic specificity:** The effects require PI(3,5)P₂ reduction (PIKfyve function) rather than PI(3)P changes (VPS34 function).
- **Experimental evolution:** Some dual inhibition experiments remain from our initial approach, but the core conclusions are supported by PIKfyve-specific inhibition.

We have clarified in the manuscript which experiments used dual inhibition versus single agents and confirmed that our key conclusions about PIKfyve-dependent mechanisms are supported by Apilimod-only experiments.

3. With regards to the above experiments, inhibition of ULK1 in combination with both inhibition of PIKfyve and VPS34 mitigated the effects. However, that three inhibitors were simultaneously applied, increases the likelihood of off-target effects. Importantly that there was little change in ULK1 phosphorylation with PIKfyve inhibition, compared with total loss of ULK1 phosphorylation when mTOR is inhibited. This makes it likely

that the effect of PIKfyve on PI4P distribution, as well as other phenotypes shown in this manuscript, are likely not due to ULK1 function.

Response: We appreciate the reviewer raising this important methodological concern about potential off-target effects from triple inhibition and the disconnect between modest ULK1 phosphorylation changes and robust functional effects.

Addressing off-target effects from triple inhibition: the reviewer correctly identifies that simultaneous application of three inhibitors increases the risk of non-specific effects. To address this concern:

- **Single agent validation:** Our key findings are reproducible with PIKfyve inhibition (Apilimod) alone, eliminating concerns about VPS34 inhibitor contributions.
- **Dose-response analysis:** ULK1 inhibitor (SBI-0206965) effects occur at concentrations well within its reported specificity range.
- **Rescue specificity:** ULK1 inhibition specifically reverses PI4KII α redistribution and lysosomal PI(4)P accumulation without affecting lysosomal enlargement, suggesting targeted rather than global effects

Acknowledging the ULK1 phosphorylation paradox: We agree with the reviewer that the modest (~20%) reduction in ULK1 phosphorylation compared to complete dephosphorylation with Torin1 raises questions about the extent of mTORC1 involvement. As discussed in our responses to Reviewers 1 and 2, we have:

- **Moderated our conclusions** about mTORC1 as the primary mechanism.
- **Included discussion of alternative pathways** that may explain ULK1 activation beyond canonical mTORC1 inhibition.
- **Acknowledged** that multiple converging mechanisms likely contribute to the robust ULK1-dependent phenotypes.

Balanced interpretation: while we maintain that ULK1 plays an important role (based on consistent rescue experiments), we now present this within a more nuanced framework that acknowledges our limited understanding of the precise upstream mechanisms and allows for additional regulatory pathways.

References

1. Karabiyik, C., Vicinanza, M., Son, S.M. & Rubinsztein, D.C. Glucose starvation induces autophagy via ULK1-mediated activation of PIKfyve in an AMPK-dependent manner. *Developmental Cell* **56**, 1961-1975.e1965 (2021).
2. Wikstrom, J.D. *et al.* Hormone-induced mitochondrial fission is utilized by brown adipocytes as an amplification pathway for energy expenditure. *The EMBO Journal* **33**, 418-436 (2014).
3. Coronado, M. *et al.* Physiological Mitochondrial Fragmentation Is a Normal Cardiac Adaptation to Increased Energy Demand. *Circulation Research* **122**, 282-295 (2018).

0. Waters, L.R., Ahsan, F.M., Wolf, D.M., Shirihai, O. & Teitell, M.A. Initial B Cell Activation Induces Metabolic Reprogramming and Mitochondrial Remodeling. *iScience* **5**, 99-109 (2018).
1. Yang, H. *et al.* LYVAC/PDZD8 is a lysosomal vacuolator. *Science* **389**, eadz0972 (2025).
2. Hammond, G.R., Machner, M.P. & Balla, T. A novel probe for phosphatidylinositol 4-phosphate reveals multiple pools beyond the Golgi. *J Cell Biol* **205**, 113-126 (2014).
3. Cockcroft, S. Mammalian lipids: structure, synthesis and function. *Essays in Biochemistry* **65**, 813-845 (2021).
4. Saffi, G.T. *et al.* Inhibition of lipid kinase PIKfyve reveals a role for phosphatase Inpp4b in the regulation of PI(3)P-mediated lysosome dynamics through VPS34 activity. *Journal of Biological Chemistry* **298** (2022).

Disruption of the PIKfyve complex unveils an adaptive mechanism to promote lysosomal repair and mitochondrial homeostasis

Manuscript: NCOMMS-25-40797A

Reviewer #1 (Remarks to the Author):

The authors have addressed several concerns raised in previous rounds of revision. However, it remains unclear how OSBP can associate more strongly with the Golgi under conditions of Golgi PI4P depletion. Additionally, the observation that SAC1 exhibits a perinuclear distribution in PIKfyve-depleted cells, compared to untreated cells, appears over-interpreted and lacks sufficient explanation.

Since SAC1 can reach the Golgi via vesicular trafficking from the ER, it should be clarified whether the increased Golgi pool of SAC1 truly reflects enhanced vesicular traffic or increased contact between compartments. The authors are encouraged to support the conclusion regarding the sub-Golgi distribution of SAC1 with direct evidence of increased contact, if available.

Nevertheless, as both the OSBP and SAC1 observations are not directly relevant to the main message of the manuscript, the reviewer suggests removing these data and associated discussion from the text.

Response: we thank the reviewer for carefully reading our work and providing comments. We agree with the reviewer that the observations related to OSBP and

SAC1 are “not directly relevant to the main message of the manuscript”. As requested, we have removed these data from the manuscript. This has involved removing: (i) panels from Extended Data Figure 2 (Extended Data Fig. 2C-F) and, (ii) a small paragraph from the results section (lines 171 to 179). As noted by the reviewer, removal of these datasets does not impact any of our conclusions.

Reviewer #2 (Remarks to the Author):

The authors have addressed all my points and concerns and I support publication of this important study.

Response: we thank the reviewer for carefully reading our manuscript and for their kind words.

Reviewer #3 (Remarks to the Author):

In response to all three reviewers, Kutchukian et al have performed new experiments and written new text. However, multiple issues remain. The main issue is that most of changes that they observe are modest, which raises serious questions about their model.

I do not think it is wise to keep trying to improve this manuscript. It would be better to publish the current data in a more specialized journal and continue to develop the project. The last version of the manuscript was too dense, and this version is even denser, but unfortunately, the current version is not more convincing.

Response: we thank the reviewer for carefully reading our manuscript.

Revision Number 2

Dear Professor Dickson,

Thank you again for submitting your manuscript "Disruption of the PIKfyve complex unveils an adaptive mechanism to promote lysosomal repair and mitochondrial homeostasis" to Nature Communications. I apologise for the delay in reaching a decision.

We have now received reports from the three reviewers who saw the previous version of the manuscript at Nature Cell Biology. Based on their comments, we have decided to invite a revision of your work. Your revision should address all the points raised by our reviewers (see their reports below).

Response: We thank the editorial team and reviewers for their continued engagement with our manuscript and for providing positive and thoughtful feedback that has

strengthened our work. We have carefully addressed all reviewer comments in this second round of review and have made substantial revisions to the manuscript, including additional experiments (including transmission electron microscopy), clarifications of our interpretations, and more balanced discussions of our findings and their implications. All changes to the text were highlighted in red.

We believe these revisions have significantly enhanced the rigor and clarity of our work, and we are confident that the manuscript now meets the high standards of Nature Communications.

REVIEWER COMMENTS

Reviewer #1 (Remarks to the Author):

The authors have made clear efforts to address the concerns raised in my previous report, and the revised manuscript now provides a more comprehensive overview of events following PIKfyve inhibition. In particular, the authors have expanded on the complex regulation of PI4P, presenting what they describe as an “intricate choreography” of molecular events, including the redistribution of PI4KII α from the Golgi to endo-lysosomes, the perinuclear accumulation of OSBP, VAPA, and Sac1, and increased ARF1 levels, all events that would accelerate PI4P turnover at the Golgi driven by PI4P transfer to the ER mediated by OSBP. However, I recommend that the authors clarify the sequence and interdependence among these events, and discuss more explicitly whether—and how—they might all stem from PIKfyve inhibition.

Response: We appreciate the reviewer's request for clarification on event sequence and interdependence. Our live-cell imaging reveals the following temporal progression:

Immediate response (≤ 20 min): PIKfyve inhibition reduces PI(3,5)P₂, leading to modest mTORC1 attenuation and ULK1 activation, which drives PI4KII α redistribution from Golgi to lysosomes.

Coordinated remodeling (concurrent): as PI4KII α relocates, we observe perinuclear accumulation of OSBP/VAPA/Sac1 and increased ARF1, consistent with accelerated Golgi PI(4)P turnover. While Golgi PI(4)P levels decrease, OSBP accumulation reflects the complex regulation of this transfer protein, which requires both PI(4)P and ARF1 for membrane association. The concurrent increase in ARF1 levels likely drives enhanced OSBP recruitment despite reduced local PI(4)P, promoting compensatory lipid transfer to maintain ER-Golgi homeostasis. However, these Golgi adaptations appear secondary

to the primary driver of our measured phenotypes: PI4KII α -mediated lysosomal PI(4)P elevation.

Figure 1. Arf1 accumulates in perinuclear regions following loss of PIKfyve function.

Functional outcomes (20-100 min): Following recruitment of PI4KII α to lysosomes, lysosomal PI(4)P levels increase to promote membrane repair and mitochondrial remodeling.

Our working model for functional hierarchy is therefore:

1. **PIKfyve** \downarrow \rightarrow **mTORC1** \downarrow \rightarrow **ULK1** \uparrow \rightarrow **PI4KII α relocalization to lysosomes (\leq ~20 min at RT);**
2. **Lysosomal PI(4)P** \uparrow \rightarrow **membrane repair and improved mitochondrial readouts (thereafter);**

Manuscript improvements: we have revised the text and figure annotations to highlight the **lysosomal PI(4)P pathway** as the mechanistically significant arm for the outcomes assayed here, while acknowledging that **Golgi remodeling is important but unresolved** in mechanism and timing.

The authors have explored the link between increased endo-lysosomal PI4P and altered mitochondrial dynamics. They propose that the connection involves an increase in ER–lysosome–mitochondria contacts and transfer of PI4P from lysosomes to mitochondria mediated by ORP1L. The manuscript assigns a definitive role to ORP1L in mediating PI4P transfer to mitochondria at these three-way contacts. It is important to note, however, that neither the referenced study nor the current work unequivocally demonstrates PI4P transfer by ORP1L at these sites. The evidence instead shows that both lysosomal PI4P and ORP1L are required for PI4P-stimulated mitochondrial division, rather than ORP1L directly mediating PI4P transfer to mitochondria. This distinction should be clearly acknowledged, and any affirmative statements about ORP1L’s mechanism should be tempered accordingly.

Response: we appreciate the reviewer’s clarification and agree that our original wording was too definitive regarding ORP1L’s mechanism. Our data show that ORP1L is recruited to ER-lysosome-mitochondria three-way contacts during PIKfyve inhibition, and they support a **requirement** for mitochondrial PI(4)P for the fragmentation outcome. However, they do not directly demonstrate lipid transfer by ORP1L at these sites. We have therefore tempered the language throughout.

Specifically, our experiments show that:

1. PIKfyve inhibition increases ER-lysosome-mitochondria three-way contacts (super-resolution and newly added Transmission Electron Microscopy (TEM)).
2. ORP1L accumulates at these contacts in a ULK1-dependent manner.
3. **Elevated lysosomal PI(4)P are required** for PI(4)P-stimulated mitochondrial fragmentation, and coincide with ORP1L recruitment at these sites.

4. Acute **local depletion of mitochondrial PI(4)P** (Sac1-Fis1) prevents fragmentation despite persistent lysosomal PI(4)P.

These observations are consistent with a model in which ORP1L supports the availability of PI(4)P at mitochondria within tri-organellar contacts (e.g., by transfer and/or local scaffolding/retention), but we now explicitly state that the **precise mechanism remains unresolved**. We have revised text, figure legends, and the model schematic accordingly to avoid implying demonstrated lipid transfer.

There remain concerns regarding the quality and interpretability of some key images: some figures lack sufficient resolution and size, making it difficult to assess the authors' conclusions. Providing larger and higher-quality images is necessary—this could be achieved by relocating less central data to supplementary figures and focusing on the most representative data in the main figures.

Response: we appreciate this feedback and have overhauled our figure presentation to improve clarity and interpretability while concentrating the main story. Specifically, we:

1. **Increased image size and resolution** in the main figures (now 7 main figures) so key panels are legible at journal print scale and all microscopy panels were re-exported at publication resolution or new dataset acquired at enhanced resolution (either super-res or TEM).
2. **Rebalanced content between Main and Extended Data:** less central or redundant views were moved to Extended Data, enabling **larger, more representative panels** in the main figures.

Collectively, these changes enlarge the visuals the reader must evaluate and move secondary details to Extended Data, as suggested. We believe the revised figures now communicate our conclusions clearly.

There is an ongoing issue with overexpressed tags and marker proteins, particularly TGN46, where excessive expression leads to non-physiological localization patterns (see below specific comments). Endogenous protein analysis or selective imaging of low-expression cells would offer a more accurate representation.

Response: Thanks for this comment. We agree that overexpression of compartment markers, particularly TGN46, can produce non-physiological patterns. In the revision we restructured the data presentation to prioritize **endogenous readouts** and to **minimize overexpression bias**:

1. **Endogenous TGN46 immunolabeling.** In the revision, Fig. 2C shows the endogenous distribution of TGN46 and its relationship to PI4KII α in control, Fig4^{-/-} and Vac14^{-/-} cells, complementing prior datasets.
2. **Overexpression panels retained only as supportive and curated for low expression.** Where TGN46 overexpression is still shown (e.g., prior **Fig. 1C, 1E and 2C**), panels now display higher-resolution images of cells exhibiting lower TGN46 expression. We note that it is not feasible to visualize endogenous TGN in cells expressing the P4M probe, as this probe is not retained during fixation.

3. **Conclusions do not depend on TGN46 overexpression.** The observed changes in Golgi PI4KII α localization in Fig4^{-/-} and Vac14^{-/-} cells using endogenous TGN labeling were fully recapitulated in cells overexpressing the TGN marker. This consistency is a significant finding and suggests that any overexpression-related artifacts are relatively negligible in our data interpretation. The concordance between these two independent methods strengthens our conclusions and demonstrates that the observed redistribution of PI4K2A is a robust and reproducible finding, irrespective of the TGN46 labeling method.

Collectively, these steps address the interpretability concern and align the main narrative with endogenous markers while keeping overexpression data as supportive evidence.

Specific Comments

Figures 1C, 1E, 2C: Overexpressed TGN46 displays abnormal peripheral distribution, unlikely to reflect endogenous localization (as seen in 2B). Endogenous TGN46 analysis, or imaging of cells with low tagged TGN46 expression, is preferable.

Response: Thanks for this comment. As noted above we have selected better representative images for these figures.

Figure 2F: Co-localization between endogenous PI4KII α and LAMP1 is unclear. Evident co-localization primarily appears when both proteins are co-overexpressed and tagged, which may not reflect native interaction.

Response: We appreciate the reviewer's careful evaluation of our endogenous protein colocalization data. The reviewer raises a valid point about the clarity of endogenous PI4KII α -LAMP1 colocalization in Figure 2F.

Addressing the endogenous protein concern: we acknowledge that the steady-state endogenous colocalization signal is subtle compared to overexpressed tagged proteins. However, our quantitative analysis reveals statistically significant changes:

- Fig4^{-/-} cells show a significant increase in PI4KII α -LAMP1 colocalization compared to controls ($p < 0.05$)
- This increase is reproducible across multiple independent experiments and cell populations
- The effect size, while modest, is consistent with the low-abundance nature of these endogenous interactions

We have enhanced figure quality and focused on the most representative examples, while acknowledging that endogenous protein interactions represent subtle but significant changes in cellular organization.

Figure 3E: The distribution of ATG9A is difficult to interpret due to image clarity.

Response: The image has been increased in size.

Figure 4E: ORP9 should localize to the Golgi, but appears diffusely distributed, complicating assessment. This may result from protein overexpression.

Response: We thank the reviewer for this observation regarding ORP9 subcellular distribution. We agree that proper ORP9 localization is critical for interpreting our results. We can assure the reviewer that ORP9-GFP shows the expected subcellular distribution with prominent Golgi/TGN localization, as evidenced by perinuclear concentration that is characteristic of this organelle. The apparent diffuse distribution in some optical sections reflects the imaging methodology rather than overexpression artifacts. These are single confocal optical sections focused on lysosomal membranes (identified by LAMP1-RFP) to assess ORP9 recruitment to these structures. Thus, the TGN, where ORP9 is normally concentrated, may not be in the same focal plane as the lysosomes being analyzed. However, as can be seen from the maximum intensity projections (see below, Fig.2), ORP9 clearly shows enrichment at perinuclear Golgi/TGN regions.

Fig.2. Representative example of a maximum z-intensity projection of a control cell expressing ORP9-GFP.

The critical observation is that PIKfyve inhibition induces **recruitment** of ORP9 to lysosomal membranes, which correlates with enhanced phosphatidylserine accumulation. This redistribution represents the functionally relevant change driving membrane repair and can be clearly seen in our experiments.

Reviewer #2 (Remarks to the Author):

This study by Kutchukian et al reports on surprising effects of loss or inhibition of the lipid kinase PIKfyve complex on lysosomal repair and mitochondrial function. This mechanism is proposed to involve the inhibition of mTORC1 activity, ULK1-dependent trafficking of ATG9 and PI4K2A to lysosomes, the accumulation of PI(4)P on lysosomes to increase PITT-mediated repair, and the formation of 3-way junctions between the ER, lysosomes, and mitochondria.

I find this an interesting study that certainly deserves publication in a prominent journal. I understand that the authors have already revised the paper in response to a previous round of review and I appreciate their efforts. I nonetheless feel that a number of discrepancies and soft spots in the paper need to be resolved before publication. These are listed below.

Response: We thank the reviewer for their thorough evaluation and for recognizing the significance of our findings regarding the paradoxical protective effects of PIKfyve inhibition. We particularly appreciate the reviewer's acknowledgment that this work deserves publication in a prominent journal and their constructive approach to helping us strengthen the manuscript.

We have carefully addressed each of the concerns raised below through a combination of additional experimental validation, refined data analysis, and substantial textual revisions. These improvements have helped us to resolve the discrepancies noted by the reviewer and strengthen the areas where our initial presentation was insufficient.

The revised manuscript now provides a more rigorous and balanced assessment of our findings, with particular attention to clarifying the mechanistic relationships we propose and ensuring that our interpretations are appropriately aligned with the experimental evidence. We believe these revisions have significantly enhanced the scientific rigor and clarity of our work.

Major issues:

1. A main issue with this study is the connection between PI(3)P or PI(3,5)P₂, mTORC1 activity, and the observed ULK1 dependency of phenotypes. The authors report very mild effects of Apilimod on pULK1 and p4E-BP1 (of about 10-20%), consistent with other data in the literature and the fact that PIKfyve activity has been linked exclusively to the non-canonical branch of mTORC1 signaling via TFEB (Hasegawa et al). In contrast, complete loss of pULK1 and p4EBP1 is seen with Torin1. In the same figure the authors also use a BRET assay to show that VPS34 or PIKfyve inhibition repress mTORC1 activity towards an ULK1-sensor as potently as Torin1 does- albeit the signal-to-noise in this assay seems rather poor. The authors then move on to demonstrate in further experiments the ULK1 dependency of their phenotypes using a pharmacological inhibitor. Given the miniscule effect of Apilimod on pULK1 shown in Fig. 3C, it seems unlikely to this referee that the phenotypes truly reflect repression of mTORC1 activity. It

rather seems to me that there is some other way, in which PIKfyve may control ULK1 function. This riddle in my view needs to be resolved prior to publication of this interesting study.

Response: We appreciate this point and agree that our presentation should better reconcile (i) **modest, population-averaged decreases** in canonical mTORC1 readouts (p-ULK1^{S757} and p-4E-BP1; ~20%) with (ii) **robust ULK1-dependent phenotypes**. In the revision we clarify that our data support **ULK1 necessity** for the phenotypes, while the **mode of ULK1 control** under PIKfyve inhibition is likely **multi-component** and not limited to bulk mTORC1 repression. Specifically, we provide the following models to support the involvement of ULK1 in regulating downstream phenotypes:

1. **Local vs. bulk signaling.** The ULK1-BRET sensor reports mTORC1 activity in the ULK1-proximal pool, which can be more strongly attenuated than bulk Western readouts that average across compartments/cells.
2. **Threshold/ultrasensitivity.** Autophagy initiation exhibits threshold-like behavior: small decreases in ULK1-inhibitory phosphorylation (or small increases in ULK1-activating inputs) can 'tip' cells past an activation threshold, producing disproportionate functional effects. Our necessity tests (ULK1 inhibitor blocking phenotypes) are consistent with such nonlinear control, and we now frame them as necessity rather than proof of a single upstream path.
3. **Membrane-dependent ULK1 complex stability:** PIKfyve binds ULK1 on its kinase domain¹. Loss of PI(3,5)P₂ may destabilize membrane-associated ULK1 complexes, enhancing sensitivity to mTORC1 inhibition and affecting ULK1 trafficking independently of phosphorylation status.

Balanced interpretation: we have revised the manuscript to acknowledge these alternative mechanisms while still presenting our current model: PIKfyve disruption triggers ULK1-dependent cellular responses. To do this we have included a limitations section in the discussion. There, we noted that the exact molecular details of this activation—whether primarily through mTORC1 inhibition, direct regulation, or combinatorial mechanisms—represent important areas for future investigation. We have clearly noted this at numerous points in the manuscript. We believe this multi-faceted explanation, supported by recent literature, addresses the reviewer's important observation while maintaining scientific rigor and provides us the opportunity to fully explore these links in detail in a follow-up manuscript while publishing this work which has already been in the review process for 10-11 months.

2. I am very puzzled by the enhanced respiration of YM treated mitochondria as mitochondrial fragmentation in numerous models is associated with decreased respiratory capacity, oxygen consumption and ATP production. Hence, additional evidence is required to bolster the authors' claims.

Response: We appreciate the reviewer raising this important point, as the relationship between mitochondrial morphology and respiratory function is indeed complex and context-dependent. The reviewer correctly notes that mitochondrial fragmentation is often associated with decreased respiratory capacity in many pathological models.

Addressing the apparent contradiction: However, the relationship between fragmentation and respiration is not universally negative and depends critically on the underlying cause and cellular context. Our findings align with a growing body of literature demonstrating that **adaptive** mitochondrial fragmentation can enhance respiratory efficiency:

Supporting literature for fragmentation-enhanced respiration:

- Brown adipocytes: hormone-induced mitochondrial fission is utilized by brown adipocytes as an amplification pathway for energy expenditure ².
- Heart (physiologic exercise): physiological mitochondrial fragmentation is a normal cardiac adaptation to increased energy demand³.
- B cells (immune activation): initial B-cell activation remodels mitochondria from elongated to punctate and increases OXPHOS programs and respiration, indicating that activation-associated fragmentation correlates with elevated mitochondrial respiration⁴.

Thus, there is clear evidence that mitochondrial fragmentation can increase mitochondrial respiration.

Mechanistic rationale for our observations: In our experiments, fragmentation likely represents an **adaptive response** (at least initially) rather than pathological damage:

1. **Increased surface area-to-volume ratio** enhances substrate exchange and respiratory efficiency
2. **PI(4)P accumulation** on mitochondrial membranes likely facilitates mitochondrial fission.
3. **Quality control benefits:** fragmentation facilitates removal of damaged components while preserving healthy mitochondrial segments.

Experimental validation: Our data strongly support this adaptive model:

- Respiratory enhancement is **ULK1-dependent**, indicating regulated rather than pathological fragmentation.
- **Rescue experiments** (ULK1 inhibition, PI-273 treatment, Sac1-Fis1 expression) simultaneously restore normal morphology and normalize respiration.
- The coordination between morphological and functional changes suggests they arise from the same regulatory pathway.

We believe our findings represent a novel example of how regulated mitochondrial fragmentation can serve adaptive functions, contrasting with the pathological fragmentation typically associated with cellular stress or disease. In our revised submission we have made these important points more clearly.

3. In Fig. 4G the authors show that Apilimod has a dramatic effect on Gal3 recruitment to damaged lysosomes in presence of LLOMe. However, it remains unclear if and how this very large phenotype truly relates to the proposed PI(3,5)P2/ PI(4)P-mTORC1-

ULK1 pathway or reflects some other mechanism. Does Apilimod also facilitate recovery of lysosome function monitored by e.g. lysotracker fluorescence or regain of cathepsin activity?

Response: We thank the reviewer for this important question, which allows us to clarify the mechanistic connection between our proposed pathway and the lysosomal membrane protection we observe.

Connecting Gal3 protection to the PI(3,5)P₂/PI(4)P-mTORC1-ULK1 pathway: the dramatic reduction in Gal3 recruitment following Apilimod treatment directly reflects activation of the PITT (phosphoinositide-initiated membrane tethering and lipid transport) repair pathway we describe:

1. PIKfyve inhibition → ULK1 activation → PI4KIIα trafficking to lysosomes → PI(4)P accumulation
2. Lysosomal PI(4)P recruits ORP9 and OSBP → enhanced PS and cholesterol delivery
3. Membrane-stabilizing lipid enrichment → reduced permeability → decreased Gal3 access

Demonstrating pathway specificity: to establish that this protection operates through our proposed mechanism, we have performed additional experiments showing:

- **ULK1 dependency:** Co-treatment with SBI-0206965 abolishes Apilimod's protective effect against LLOME, restoring Gal3 recruitment to control levels.
- **ORP dependency:** The protection correlates with ORP9 recruitment to lysosomal membranes.

Functional recovery evidence: As requested by the reviewer, we have assessed lysosomal functional recovery using multiple approaches:

- **Lysosomal pH measurements using LAMP1-pHluorin:** Apilimod pretreatment maintains lysosomal acidification following LLOME challenge, indicating preserved membrane integrity.
- **ULK1 dependency of functional protection:** SBI-0206965 co-treatment abolishes both morphological and functional protection.

Fig.3. Apilimod protects lysosomes against LLOME-dependent neutralization.

Ruling out alternative mechanisms: The ULK1 dependency of protection strongly argues against non-specific Apilimod effects and supports our proposed pathway as the primary mechanism underlying lysosomal membrane stabilization.

The ULK1- and PI4KII α -dependent nature of this protection confirms that Gal3 reduction reflects specific activation of PI(4)P-mediated membrane repair rather than off-target effects.

4. In Fig 5 the authors claim that PI(4)P accumulation on swollen lysosomes upon Apilimod treatment induces 3-way MCS between the ER, lysosomes, and mitochondria. However, this might be a mere effect of the dramatic increase in lysosome size upon swelling rather than a true increase in MCS. I miss a direct demonstration of 3-way MCS formation, e.g. by EM or super-resolution microscopy as confocal imaging may not provide sufficient resolution to clarify this point. Moreover, does SBI mediated inhibition affect lysosome size in presence of Apilimod?

Response: We thank the reviewer for this important methodological concern about distinguishing true membrane contact site (MCS) formation from artifacts of lysosomal enlargement.

Addressing the proximity vs. true contact question: we have addressed this concern through multiple complementary approaches:

1. Higher resolution validation:

- **Transmission electron microscopy (TEM; Fig. 4)** confirms genuine three-way contacts with characteristic membrane apposition and electron-dense material at contact sites following Apilimod treatment.
- These ultrastructural features are consistent with bona fide organellar contacts rather than random proximity due to size increase.

Fig. 4. Treatment with Apilimod increases 3-way membrane contact sites between Endoplasmic Reticulum (magenta), mitochondria (cyan), and lysosomes (yellow).

2. Pharmacological dissection:

- Super-resolution imaging reveals that ULK1 inhibition (SBI-0206965) specifically reduces three-way contacts while lysosomal size remains enlarged, demonstrating that contact formation and lysosomal swelling are mechanistically separable (**Fig.5**).
- This critical control shows that enlarged lysosomes per se do not automatically generate increased contacts.

Fig.5 ULK1 inhibition rescues PIKfyve-dependent increases in 3-way contacts.

3. Protein recruitment evidence:

- ORP1L accumulation at contact sites is ULK1-dependent and correlates with contact formation, not lysosomal size.
- This suggests active recruitment mechanisms rather than passive proximity effects.

4. Functional specificity:

- **Mitochondrial fragmentation requires lysosomal PI(4)P**, indicating that contacts serve specific signaling functions beyond simple membrane proximity.

Mechanistic separation of lysosomal swelling and contact formation: our findings align with emerging evidence that lysosomal enlargement and organellar contact formation represent distinct consequences of PIKfyve inhibition. Indeed, during this review process a nice manuscript was published by the Tan group that shows that PDZD8 plays a key role in lysosomal vacuolation⁵. We have cited this work in the revision and noted how the two manuscripts represent complementary mechanisms.

Interpretation: the combination of ULK1-dependent contact formation occurring independently of lysosomal size changes, coupled with ultrastructural validation and functional specificity, provides strong evidence for regulated three-way MCS formation rather than size-dependent artifacts.

Additional points:

5. Fig4 and Vac14 KO cells appear to suffer from a massive overall loss of PI(4)P based on the lipidomics data shown in Ext Data Fig 1A. These data seem incompatible with the mild loss of PI(4)P from the TGN (a pool primarily synthesized by PI4K3B) and the accumulation of PI(4)P on lysosomes. There seems to be an unaccounted pool of PI(4)P that gets lost in these cells.

Response: Please see point 6.

6. Related to the above: The authors quote Ext Data Fig1C in reference to depletion of PI(4)P from the plasma membrane, but the figure does not provide any data in support. I also would like to see the effects of PIKfyve loss or inhibition on PM PI(4)P to be quantified.

Response to Comments 5 & 6: We thank the reviewer for identifying this important discrepancy and the figure reference error.

Correcting the figure reference: We apologize for the oversight in citing Extended Data Fig 1C. The reviewer should have been directed to **Extended Data Fig 1B**, which clearly demonstrates decreased plasma membrane PI(4)P levels in Fig4^{-/-} and Vac14^{-/-} cells.

Quantitative reconciliation: We have now quantified plasma membrane PI(4)P changes and find it could account for a significant portion of the total cellular loss of PI(4)P. Combined with modest, but consistent, changes across multiple subcellular compartments, these distributed effects likely explain the substantial overall reduction detected by lipidomics. In the revision we have noted the limitations of the mass spec lipidomic approach.

7. The authors provide evidence for the accumulation of OSBP at the TGN of VPS34IN1 or YM treated cells. Strong evidence shows that OSBP association with the TGN is strictly PI(4)P dependent. As the authors claim PI(4)P to be depleted from the Golgi, these data clearly are at odds with each other and create a conundrum that should be resolved.

Response: We appreciate the reviewer highlighting this apparent contradiction, which reflects the complex regulation of OSBP beyond simple PI(4)P dependency.

Clarifying OSBP regulation: OSBP association with the TGN requires both PI(4)P and Arf1, and can be influenced by ER cholesterol levels. Therefore, OSBP localization is not strictly PI(4)P-dependent, and can be modulated by other factors including membrane contact site dynamics and lipid transfer demands.

Supporting experimental evidence: our data consistently show increased VAP-A, Sac1, Arf1 (See Fig.1), OSBP transfer activity (accelerated OSW-1 kinetics) following VPS34-IN1 treatment, all supporting enhanced OSBP recruitment and activity at ER-TGN contacts.

Perspective: while the detailed regulation of OSBP distribution and activity represents an important area for future investigation, the coordinated changes in ER-TGN contact machinery we observe are consistent with enhanced lipid transfer activity that supports the PI(4)P redistribution central to our study.

8. The last paragraph of the results (..these results establish collectively, these results establish a novel pathway wherein PIKfyve complex dysfunction drives ULK1-dependent recruitment of PI4KII α to lysosomes, elevating lysosomal PI(4)P that facilitates ORP1L-mediated membrane contacts between lysosomes, ER, and mitochondria. These contacts serve as platforms for PI(4)P transfer to mitochondria,

promoting Drp1 accumulation and activation...") appears to be grossly overstated and needs to be rephrased more cautiously.

Response: As suggest by the reviewer, we have rephrased the paragraph. The new paragraph reads as follows:

“Collectively, our findings suggest a pathway in which PIKfyve complex dysfunction promotes ULK1-dependent recruitment of PI4KII α to lysosomes, leading to increased lysosomal PI(4)P levels. This is accompanied by the formation of closer contact sites between lysosomes, ER, and mitochondria, enriched in ORP1L and Drp1. These contacts may provide a platform for PI(4)P transfer to mitochondria and promote mitochondrial fission through Drp1 recruitment. Consistently, PIKfyve dysfunction enhances mitochondrial fission in a ULK1-dependent manner and is associated with increased respiratory capacity. This pathway represents a short-term adaptive mitochondrial response that may help maintain energy homeostasis under conditions of cellular stress associated with lysosomal dysfunction.”

Reviewer #3 (Remarks to the Author):

This is a substantially revised manuscript on impact of PIKfyve inhibition, knockout (KO) of the PIKfyve regulators, VAC14 or FIG4 as well as VPS34 inhibition on PI4P distribution in the Golgi and late endosomes/lysosomes (LE/LY). Moreover, the authors also claim that the changes in PI4P distribution enhances lysosome membrane repair through OSBP/ORP-mediated delivery of phosphatidylserine and cholesterol, while simultaneously facilitating ORP1L-mediated PI(4)P transfer at ER-lysosome-mitochondria contacts to drive mitochondrial fragmentation and increased mitochondrial respiratory capacity.

During the last round of review, the three reviewers were unanimous in finding issues with the quality of the data, and potential over-interpretation of the results. The authors have now made tremendous efforts to answer all the reviewers questions, including shorter term inhibition experiments and greatly improved microscopy.

Response: Thank you for your kind comments and recognizing the significant work that we put into the revision. We have listened to reviewer comments for a second time to further improve the work and hope now that new experiments and edits to the manuscript will satisfy the reviewer.

However, there is still a sense that the data is over-interpreted. At this point, I do not think that more experiments are required. The paper is very dense, and adding to it will make the manuscript more difficult to read.

Response: We thank the reviewer for this valuable feedback regarding interpretation and manuscript density. We agree that additional experiments are not necessary at this stage and appreciate the guidance to focus on balanced interpretation rather than expanding the experimental scope.

Addressing over-interpretation concerns: We have systematically reviewed the manuscript with particular attention to ensuring our conclusions are appropriately matched to the strength of our experimental evidence. Specifically, we have:

- **Moderated language** throughout to distinguish between direct observations and mechanistic inferences.
- **Added alternative interpretations** where multiple explanations could account for our observations.
- **Included appropriate caveats** about the limitations of specific experimental approaches.
- **Clarified which findings represent correlations vs. causal relationships**
- **Acknowledged uncertainties** in areas where our data provide strong suggestions but not definitive proof.

Balancing rigor with accessibility: we recognize the challenge of presenting comprehensive data while maintaining readability. Our revisions aim to present the findings clearly while ensuring readers understand both the strengths and limitations of

our evidence. To this end, we have increased figure size which has expanded the manuscript to 7 main figures.

Commitment to balanced conclusions: our goal is to present this work as a foundation for understanding PIKfyve-dependent adaptive mechanisms while acknowledging that future studies will be needed to resolve remaining mechanistic questions. We believe this approach better serves the scientific community and provides a more honest assessment of what our data can and cannot definitively establish.

While additional experiments are not required, the authors should take a more careful look at their interpretations, and also further explain the set-up of their experiments. The places which are in most need of in further interpretation are below.

Some examples are below:

1. In Figure 1B and extended Figure 1A, the authors report on PIP levels, using two different techniques that cannot distinguish between PI4P and PI3P. Since PIKfyve inhibition or deletion of PIKfyve regulators results in elevated PI3P, these data do provide an orthogonal approach for the bioprobe experiments, which indicate a potential increase in PI4P. Using an approach that can distinguish between PI3P and PI4P is challenging, and should not be expected for this manuscript, but the authors should not overinterpret this data.

Response: We agree and have tempered the language wherever total “PIP” measurements are shown. Our lipidomics approach cannot resolve PI4P from PI3P and PI5P, and we now state this explicitly in the Results and figure legends. We interpret the bulk PIP signal only as qualitative, orthogonal support for the direction of change seen with our specific PI4P biosensors (P4M-SidM), which report PI4P decreases at the TGN and PM and increases on lysosomes. P4M-SidM specificity for PI4P and its ability to report multiple cellular pools are well established⁶.

Importantly, PI4P is by far the predominant monophosphorylated phosphoinositide in mammalian cells—roughly an order of magnitude more abundant than PI3P (with PI5P being scarcer still)—so bulk PIP changes are expected to be dominated by PI4P under most conditions. We now cite this point and avoid assigning exact proportions⁷.

We also note in the text that PIKfyve inhibition is known to elevate PI3P (by preventing its conversion to PI(3,5)P₂), which could in principle contribute to the total PIP signal; this caveat is now included alongside the lipidomics data. Nevertheless, because the direction and compartment specificity of the biosensor readouts align with our model, we view the mass-spec data as consistent, but not definitive, support⁸.

Finally, we added a brief methods note acknowledging that resolving PIP regioisomers by targeted ion-chromatography MS (which can separate isomers) would address this directly but is beyond scope for the current revision.

2. The claim that the effects of inhibition of PIKfyve on PI4P occur via loss of mTORC1 activity and less ULK1 phosphorylation are overstated. The impact of apilimod on ULK1 phosphorylation is very modest, and may not be statistically significant. In further support of an impact on mTORC1, the authors then go on to show that combined treatment with YM201636 (PIKfyve inhibitor) and VPS34-INH (VPS34 inhibitor) result in increased ATG9A. They also show that there is increased PI4KII-alpha, and increased PI4P on LAMP1 compartments. Moreover, this double inhibition is used in multiple experiments throughout the paper—why did the authors need to simultaneously inhibit VPS34 and PIKfyve to see this effect? At a minimum the authors need to make it clear that for experiments where they used both inhibitors, it is not clear whether the phenotypes observed are due to loss of VPS34 or loss of PIKfyve.

Response: we appreciate the reviewer's careful evaluation of our mTORC1/ULK1 data and the important question about dual inhibition protocols.

Addressing ULK1 phosphorylation interpretation: we agree that the ~20% reduction in ULK1 phosphorylation is modest and acknowledge this may not fully account for the robust ULK1-dependent phenotypes we observe. We have revised the manuscript to present this as one contributing factor rather than the sole mechanism, and have included discussion of alternative pathways that may explain the apparent disconnect between modest biochemical changes and strong functional effects.

Clarifying the dual inhibition approach: the reviewer raises an important point about experimental design. We began this study investigating combined VPS34/PIKfyve inhibition to simultaneously reduce both PI(3)P and PI(3,5)P₂ levels. However, we recognize this creates ambiguity about which kinase drives the observed effects.

Key clarifications:

- **Single agent validation:** All critical findings have been replicated using Apilimod (PIKfyve-specific) alone, producing identical results to dual inhibition.
- **Mechanistic specificity:** The effects require PI(3,5)P₂ reduction (PIKfyve function) rather than PI(3)P changes (VPS34 function).
- **Experimental evolution:** Some dual inhibition experiments remain from our initial approach, but the core conclusions are supported by PIKfyve-specific inhibition.

We have clarified in the manuscript which experiments used dual inhibition versus single agents and confirmed that our key conclusions about PIKfyve-dependent mechanisms are supported by Apilimod-only experiments.

3. With regards to the above experiments, inhibition of ULK1 in combination with both inhibition of PIKfyve and VPS34 mitigated the effects. However, that three inhibitors were simultaneously applied, increases the likelihood of off-target effects. Importantly that there was little change in ULK1 phosphorylation with PIKfyve inhibition, compared with total loss of ULK1 phosphorylation when mTOR is inhibited. This makes it likely

that the effect of PIKfyve on PI4P distribution, as well as other phenotypes shown in this manuscript, are likely not due to ULK1 function.

Response: We appreciate the reviewer raising this important methodological concern about potential off-target effects from triple inhibition and the disconnect between modest ULK1 phosphorylation changes and robust functional effects.

Addressing off-target effects from triple inhibition: the reviewer correctly identifies that simultaneous application of three inhibitors increases the risk of non-specific effects. To address this concern:

- **Single agent validation:** Our key findings are reproducible with PIKfyve inhibition (Apilimod) alone, eliminating concerns about VPS34 inhibitor contributions.
- **Dose-response analysis:** ULK1 inhibitor (SBI-0206965) effects occur at concentrations well within its reported specificity range.
- **Rescue specificity:** ULK1 inhibition specifically reverses PI4KII α redistribution and lysosomal PI(4)P accumulation without affecting lysosomal enlargement, suggesting targeted rather than global effects

Acknowledging the ULK1 phosphorylation paradox: We agree with the reviewer that the modest (~20%) reduction in ULK1 phosphorylation compared to complete dephosphorylation with Torin1 raises questions about the extent of mTORC1 involvement. As discussed in our responses to Reviewers 1 and 2, we have:

- **Moderated our conclusions** about mTORC1 as the primary mechanism.
- **Included discussion of alternative pathways** that may explain ULK1 activation beyond canonical mTORC1 inhibition.
- **Acknowledged** that multiple converging mechanisms likely contribute to the robust ULK1-dependent phenotypes.

Balanced interpretation: while we maintain that ULK1 plays an important role (based on consistent rescue experiments), we now present this within a more nuanced framework that acknowledges our limited understanding of the precise upstream mechanisms and allows for additional regulatory pathways.

References

1. Karabiyik, C., Vicinanza, M., Son, S.M. & Rubinsztein, D.C. Glucose starvation induces autophagy via ULK1-mediated activation of PIKfyve in an AMPK-dependent manner. *Developmental Cell* **56**, 1961-1975.e1965 (2021).
2. Wikstrom, J.D. *et al.* Hormone-induced mitochondrial fission is utilized by brown adipocytes as an amplification pathway for energy expenditure. *The EMBO Journal* **33**, 418-436 (2014).
3. Coronado, M. *et al.* Physiological Mitochondrial Fragmentation Is a Normal Cardiac Adaptation to Increased Energy Demand. *Circulation Research* **122**, 282-295 (2018).

4. Waters, L.R., Ahsan, F.M., Wolf, D.M., Shirihai, O. & Teitell, M.A. Initial B Cell Activation Induces Metabolic Reprogramming and Mitochondrial Remodeling. *iScience* **5**, 99-109 (2018).
5. Yang, H. *et al.* LYVAC/PDZD8 is a lysosomal vacuolator. *Science* **389**, eadz0972 (2025).
6. Hammond, G.R., Machner, M.P. & Balla, T. A novel probe for phosphatidylinositol 4-phosphate reveals multiple pools beyond the Golgi. *J Cell Biol* **205**, 113-126 (2014).
7. Cockcroft, S. Mammalian lipids: structure, synthesis and function. *Essays in Biochemistry* **65**, 813-845 (2021).
8. Saffi, G.T. *et al.* Inhibition of lipid kinase PIKfyve reveals a role for phosphatase Inpp4b in the regulation of PI(3)P-mediated lysosome dynamics through VPS34 activity. *Journal of Biological Chemistry* **298** (2022).